# DIFFUSION TRANSFORMERS
# WITH REPRESENTATION AUTOENCODERS

**Boyang Zheng   Nanye Ma   Shengbang Tong   Saining Xie**
New York University

## ABSTRACT

Latent generative modeling, where a pretrained autoencoder maps pixels into a latent space for the diffusion process, has become the standard strategy for Diffusion Transformers (DiT); however, the autoencoder component has barely evolved. Most DiTs continue to rely on the original VAE encoder, which introduces several limitations: outdated backbones that compromise architectural simplicity, low-dimensional latent spaces that restrict information capacity, and weak representations that result from purely reconstruction-based training and ultimately limit generative quality. In this work, we explore replacing the VAE with pretrained representation encoders (e.g., DINO, SigLIP, MAE) paired with trained decoders, forming what we term Representation Autoencoders (RAEs). These models provide both high-quality reconstructions and semantically rich latent spaces, while allowing for a scalable transformer-based architecture. Since these latent spaces are typically high-dimensional, a key challenge is enabling diffusion transformers to operate effectively within them. We analyze the sources of this difficulty, propose theoretically motivated solutions, and validate them empirically. Our approach achieves faster convergence without auxiliary representation alignment losses. Using a DiT variant equipped with a lightweight, wide DDT head, we achieve strong image generation results on ImageNet: 1.51 FID at $256 \times 256$ (no guidance) and 1.13 at both $256 \times 256$ and $512 \times 512$ (with guidance). RAE offers clear advantages and should be the new default for diffusion transformer training.

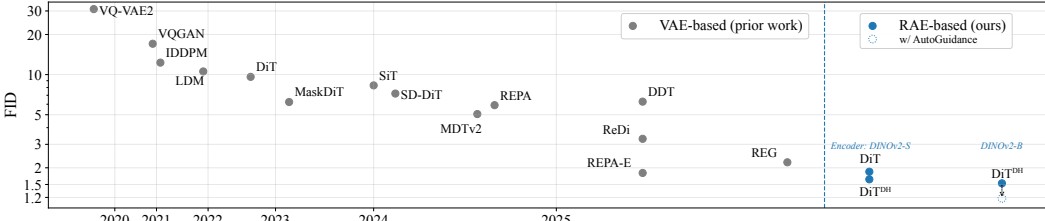

Figure 1: *Representation Autoencoder* (RAE) uses frozen pretrained representations as the encoder with a lightweight decoder to reconstruct input images without compression. RAE enables faster convergence and higher-quality samples in latent diffusion training compared to VAE-based models.

## 1 INTRODUCTION

The evolution of generative modeling has been driven by a continual redefinition of *where* and *how* models learn to represent data. Early pixel-space models sought to directly capture image statistics, but the emergence of latent diffusion (Vahdat et al., 2021; Rombach et al., 2022) reframed generation as a process operating within a learned, compact representation space. By diffusing in this space rather than in raw pixels, models such as Latent Diffusion Models (LDM) (Rombach et al., 2022) and Diffusion Transformers (DiT) (Peebles & Xie, 2023; Ma et al., 2024) achieve higher visual fidelity and efficiency, powering the most capable image and video generators of today.

Despite progress in diffusion backbones, the autoencoder defining the latent space remains largely unchanged. The widely used SD-VAE (Rombach et al., 2022) still relies on heavy channel-wise compression and a reconstruction-only objective, producing low-capacity latents that capture local appearance but lack global semantic structure crucial for generalization and generative performance of diffusion models (Song et al., 2025). In addition, SD-VAE, built on a legacy convolutional design, remains computationally inefficient (see Fig. 2).

Meanwhile, visual representation learning has undergone a rapid transformation. Self-supervised and multimodal encoders such as DINO (Oquab et al., 2023), MAE (He et al., 2021), JEPA (Assran

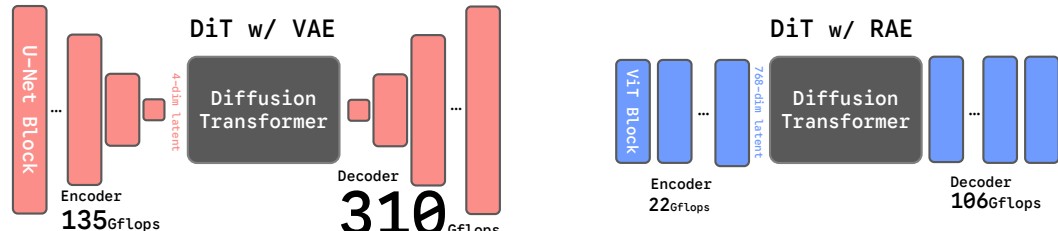

Figure 2: **Comparison of SD-VAE and RAE** (DINOv2-B). The VAE relies on convolutional backbones with aggressive down- and up-sampling, while the RAE uses a ViT architecture *without* compression. SD-VAE is also more computationally expensive, requiring about 6× and 3× more GFLOPs than RAE for the encoder and decoder, respectively. GFlops are evaluated on one $256 \times 256$ image.

et al., 2023) and CLIP / SigLIP (Radford et al., 2021; Tschannen et al., 2025) learn semantically structured visual features that generalize across tasks and scales and provide a natural basis for visual understanding. However, latent diffusion remains largely isolated from this progress, continuing to diffuse in reconstruction-trained VAE spaces rather than semantically meaningful representational ones. Recent work attempts to improve latent quality *indirectly* through REPA-style (Yu et al., 2025; Yao et al., 2025; Leng et al., 2025) alignment with external encoders, but these methods introduce extra training stages, auxiliary losses, and tuning complexity.

This separation stems from long-standing assumptions about the *incompatibility* between semantic and generative objectives. It is widely believed that encoders trained to capture semantics are *unsuited* for faithful reconstruction, since they *"focus on high-level information and can only reconstruct an image with high-level semantic similarities."* (Yu et al., 2024b) In addition, diffusion models are believed to perform poorly in high-dimensional latent spaces (Skorokhodov et al., 2025; Yao et al., 2025; Esser et al., 2024; Liu et al., 2024), leading practitioners to favor low-dimensional VAE latents over the typically much higher-dimensional representations of semantic encoders. In this work, we show that *both assumptions might be wrong*. We demonstrate that *frozen* representation encoders, even those explicitly optimized for semantics over reconstruction, can be repurposed into powerful autoencoders for generation, yielding reconstructions superior to SD-VAE without architectural complexity or auxiliary losses. Furthermore, we find that diffusion transformer training can be stable and efficient in these higher-dimensional latent spaces. With the right architectural adjustments, higher-dimensional representations are not a liability but an *advantage*, offering richer structure, faster convergence, and better generation quality. Notably, higher-dimensional latents introduce effectively *no extra compute or memory costs* since the token count is fixed (determined by the patch size) and the channels are projected to the DiT hidden dimension in the first layer.

We formalize this insight through *Representation Autoencoders* (RAEs), a new class of autoencoders that replace the VAE with a pretrained representation encoder (e.g., DINO) paired with a trained decoder. RAEs produce latent spaces that are semantically rich, structurally coherent, and *diffusion-friendly*, linking semantic and generative modeling through a shared latent representation.

While feasible in principle, adapting diffusion transformers to these high-dimensional semantic latents requires careful design. Original DiTs, designed for compact SD-VAEs, struggle with the increased dimensionality due to: (1) *Transformer design:* DiTs cannot fit even a single image unless their width exceeds the token dimension, implying width must scale with latent dimensionality; (2) *Noise scheduling:* resolution-based schedule shifts (Chen, 2023; Hoogeboom et al., 2023; Esser et al., 2024), derived from pixel- and VAE-based inputs, neglect token dimensionality, motivating a dimensionality-dependent shift; (3) *Decoder robustness:* unlike VAEs trained on continuous latent distributions (Kingma & Welling, 2014), RAE decoders learn from discretely supported latents but must reconstruct samples from a diffusion model that follow a continuous distribution, which we address by noise-augmented decoder training. Finally, we introduce a new DiT variant, DiT$^{\text{DH}}$, inspired by DDT (Wang et al., 2025c) but motivated by a different design perspective. It augments the standard DiT architecture with a lightweight, *shallow yet wide* head, enabling the diffusion model to scale in width without incurring quadratic computational costs. Empirically, this design further enhances diffusion transformer training in high-dimensional RAE spaces.

Empirically, RAEs demonstrate strong visual generation performance. On ImageNet, our RAE-based DiT$^{\text{DH}}$ achieves FIDs of 1.51 at 256×256 without any guidance, and 1.13 at both 256×256 and 512×512 with AutoGuidance (Karras et al., 2025), showing the effectiveness of RAEs as an alternative to conventional VAEs in diffusion transformer training. More broadly, these results reframes autoencoding from a compression mechanism into a representation foundation, one that enables diffusion transformers to train more efficiently and generate more effectively.

| Model | rFID |
|---|---|
| DINOv2-B | 0.49 |
| SigLIP2-B | 0.53 |
| MAE-B | 0.16 |
| SD-VAE | 0.62 |

| Decoder | rFID | GFLOPs |
|---|---|---|
| ViT-B | 0.58 | 22.2 |
| ViT-L | 0.50 | 78.1 |
| ViT-XL | 0.49 | 106.7 |
| SD-VAE | 0.62 | 310.4 |

| Encoder | rFID |
|---|---|
| DINOv2-S | 0.52 |
| DINOv2-B | 0.49 |
| DINOv2-L | 0.52 |

| Model | Top-1 Acc. |
|---|---|
| DINOv2-B | 84.5 |
| SigLIP2-B | 79.1 |
| MAE-B | 68.0 |
| SD-VAE | 8.0 |

(a) **Encoder choice.** All encoders outperform SD-VAE.

(b) **Larger decoders** improve rFID while remaining much more efficient than VAEs.

(c) **Encoder scaling.** rFID is stable across RAE sizes.

(d) **Representation quality.** RAEs have much higher linear probing accuracy than VAEs.

Table 1: RAEs consistently outperform SD-VAE in reconstruction (rFID) and representation quality (linear probing accuracy) on ImageNet-1K, while being more efficient. If not specified, we use ViT-XL as the decoder and DINOv2-B as the encoder for RAE. Default settings in this paper are in gray.

## 2 HIGH FIDELITY RECONSTRUCTION FROM FROZEN ENCODERS

In this section, we challenge the common assumption that pretrained representation encoders, such as DINOv2 (Oquab et al., 2023) and SigLIP2 (Tschannen et al., 2025), are unsuitable for the reconstruction task because they *"emphasize high-level semantics while downplaying low-level details"* (Tang et al., 2025; Yu et al., 2024b). We show that, with a properly trained decoder, frozen representation encoders can in fact serve as strong encoders for the diffusion latent space. Our **Representation Autoencoders (RAE)** pair frozen, pretrained representation encoders with a ViT-based decoder, yielding reconstructions on par with or even better than SD-VAE. More importantly, RAEs alleviate the fundamental limitations of VAEs (Kingma & Welling, 2014), whose heavily compressed latent space (e.g. SD-VAE maps $256^2$ images to $32^2 \times 4$ (Esser et al., 2021; Rombach et al., 2022) latents) restricts reconstruction fidelity and more importantly, representation quality.

Our training recipe for the RAE decoder is as follows. Given an input $\mathbf{x} \in \mathbb{R}^{3 \times H \times W}$ and the frozen representation encoder $E$ with patch size $p_e$ and hidden size $d$, we obtain $N = HW/p_e^2$ tokens with channel $d$. A ViT decoder $D$ with patch size $p_d$ maps them back to pixels with shape $3 \times H \frac{p_d}{p_e} \times W \frac{p_d}{p_e}$; By default we use $p_d = p_e$, so the reconstruction matches the input resolution. For all experiments on $256 \times 256$ images, the encoder produces 256 tokens, matching the token count of most prior DiT-based models trained with SD-VAE latents (Peebles & Xie, 2023; Yu et al., 2025; Ma et al., 2024). The decoder $D$ is trained with a combination of L1, LPIPS (Zhang et al., 2018), and adversarial losses (Goodfellow et al., 2014), following common practice in VAEs:

$$z = E(x), \hat{x} = D(z)$$
$$\mathcal{L}_{rec}(x) = \omega_L \, \text{LPIPS}(\hat{x}, x) + \text{L1}(\hat{x}, x) + \omega_G \lambda \, \text{GAN}(\hat{x}, x),$$

We provide implementation details about the decoder architecture, hyperparameters, coefficients, and GAN training details in Appendix C.

We select three representative encoders from different pretraining paradigms: DINOv2-B (Oquab et al., 2023) ($p_e$=14, $d$=768), a self-supervised self-distillation model; SigLIP2-B (Tschannen et al., 2025) ($p_e$=16, $d$=768), a language-supervised model; and MAE-B (He et al., 2021) ($p_e$=16, $d$=768), a masked autoencoder. For DINOv2, we also study different model sizes S,B,L ($d$=384,768,1024). Unless otherwise specified, we use an ViT-XL decoder for all RAEs. We use FID score (Heusel et al., 2017) computed on the reconstructed ImageNet (Russakovsky et al., 2015) validation set as our main metric for reconstruction quality, denoted as rFID.

**Reconstruction, scaling, and representation.** As shown in Table 1a, RAEs with frozen encoders achieve consistently better reconstruction quality (rFID) than SD-VAE. For instance, RAE with MAE-B/16 reaches an rFID of 0.16, clearly outperforming SD-VAE and challenging the assumption that representation encoders cannot recover pixel-level detail.

We next study the scaling behavior of both encoders and decoders. As shown in Table 1c, reconstruction quality remains stable across DINOv2-S, B, and L, indicating that even small representation encoders models preserve sufficient low-level detail for decoding. On the decoder side (Table 1b), increasing capacity consistently improves rFID: from 0.58 with ViT-B to 0.49 with ViT-XL. Importantly, ViT-B already outperforms SD-VAE while being $14\times$ more efficient in GFLOPs, and ViT-XL further improves quality at only one-third of SD-VAE's cost.

We also evaluate representation quality via linear probing on ImageNet-1K in Table 1d. Because RAEs use frozen pretrained encoders, they directly inherit the representation of the underlying representation encoders. In contrast, SD-VAE achieves only ∼8% accuracy.

Figure 3: **Overfitting to a single sample.** Left: increasing model width lead to lower loss and better sample quality; Right: changing model depth has marginal effect on overfitting results.

## 3 TAMING DIFFUSION TRANSFORMERS FOR RAE

With RAE demonstrating good reconstruction quality, we now proceed to investigate the *diffusability* (Skorokhodov et al., 2025) of its latent space; that is, how easily its latent distribution can be modeled by a diffusion model, and how good the generation performance can be.

Before turning to generation, we fix the encoder to study generation capabilities. Table 1a shows that MAE, SigLIP2, and DINOv2 all achieve lower reconstruction rFID than SD-VAE, with MAE the best among them. However, reconstruction alone does not determine generation quality. Empirically, DINOv2 produces the strongest generation results, so unless otherwise noted, we adopt DINOv2 as our default encoder and defer the full comparison to Appendix G.1.

Following standard practice, we adopt the flow matching objective (Lipman et al., 2023; Liu et al., 2023) with linear interpolation $\mathbf{x}_t = (1 - t)\mathbf{x} + t\varepsilon$, where $\mathbf{x} \sim p(\mathbf{x})$ and $\varepsilon \sim \mathcal{N}(0, \mathbf{I})$, and train the model to predict the velocity $v(\mathbf{x}_t, t)$ (see Appendix J). We use LightningDiT (Yao et al., 2025), a variant of DiT (Peebles & Xie, 2023), as our model backbone. We adopt a patch size of 1, which results in a sequence length of 256 for all RAEs on $256 \times 256$ images, matching the token length used by VAE-based DiTs (Peebles & Xie, 2023; Yu et al., 2025; Yao et al., 2025). Since the computational cost of DiT depends primarily on the sequence length, **using DiT on RAE therefore effectively incurs no additional overhead compared to its VAE-based counterparts**. We evaluate our models using FID computed on 50K samples generated with 50 steps with the Euler sampler (denoted as gFID), and all quantitative results are trained for 80 training epochs on ImageNet at $256 \times 256$ unless otherwise specified. More training details are included in Appendix D.

**DiT does not work out of the box.** To our surprise, the standard diffusion recipe fails with RAE (see Table 2). Training directly on RAE latents causes a small backbone such as DiT-S to completely fail, while a larger backbone like DiT-XL significantly underperforms it's counterpart with the SD-VAE latents.

|        | RAE    | SD-VAE    |
|--------|--------|-----------|
| DiT-S  | 215.76 | **51.74** |
| DiT-XL | 23.08  | **7.13**  |

Table 2: **Standard DiT struggles to model RAE's latent distribution.**

To investigate this observation, we raise several hypotheses detailed below, which we will discuss in the following sections:

- **Suboptimal design for diffusion transformers.** When modeling high-dimensional RAE tokens, the optimal design choices for diffusion transformers can diverge from those of the standard DiT, which was originally tailored for low-dimensional VAE tokens.

- **Suboptimal noise scheduling.** Prior noise scheduling and loss re-weighting tricks are derived for image-based or VAE-based input, and it remains unclear if they transfer well to high-dimension semantic tokens.

- **Diffusion generates noisy latents.** VAE decoders are trained to reconstruct images from noisy latents, making them more tolerant to small noises in diffusion outputs. In contrast, RAE decoders are trained on only clean latents and may therefore struggle to generalize.

### 3.1 SCALING DIT WIDTH TO MATCH TOKEN DIMENSIONALITY

To better understand the training dynamics of diffusion transformers with RAE latents, we first construct a simplified experiment. Rather than training on the entire ImageNet, we randomly select **a single image**, encode it by RAE, and test whether the diffusion model can *reconstruct* it.

Table 2 shows that although RAE underperforms SD-VAE, DiT performance improves with increased capacity. To dissect this effect, we vary model width while fixing depth. Using a DiT-S, we increase the hidden dimension from 384 to 784. As shown in Figure 3, sample quality is poor when the model width $d <$ token dimension $n = 768$, but improves sharply and reproduces the input almost perfectly once $d \geq n$. Training losses exhibit the same trend, converging only when $d \geq n$.

One might suspect that this improvement still arises from the larger model capacity. To disentangle this effect, we fix the width at $d = 384$ and vary the depth of the SiT-S model. As shown in Figure 3, even when doubling the depth from 12 to 24, the generated images remain artifact-heavy, and the training losses shown in Figure 3 fail to converge to similar level of $d = 768$.

Together, the results indicate that for generation in RAE's latent space to succeed, **the diffusion model's width must match or exceed the RAE's token dimension**. At first glance, this appears to contradict the common belief that data manifolds have low intrinsic dimensionality (Pope et al., 2021), allowing generative models such as GANs to operate effectively within that manifold without scaling to the full data dimension (Sauer et al., 2022). We argue that this contradiction arises from the formulation of diffusion models (Appendix J): injecting Gaussian noise directly to the data (e.g., in the construction of $x_t$) throughout training effectively extends the data distribution's support to the entire space, thereby "diffusing" the data manifold into a full-rank one, requiring model capacity that scales with the full data dimensionality.

In the following, we provide a theoretical justification for this conjecture:

**Theorem 1.** *Assuming* $\mathbf{x} \sim p(\mathbf{x}) \in \mathbb{R}^n$, $\boldsymbol{\varepsilon} \sim \mathcal{N}(0, \mathbf{I}_n)$, $t \in [0, 1]$. *Let* $\mathbf{x}_t = (1 - t)\mathbf{x} + t\boldsymbol{\varepsilon}$, *consider the function family*

$$\mathcal{G}_d = \{g(\mathbf{x}_t, t) = \boldsymbol{B}f(\boldsymbol{A}\mathbf{x}_t, t) : \boldsymbol{A} \in \mathbb{R}^{d \times n}, \boldsymbol{B} \in \mathbb{R}^{n \times d}, f : [0, 1] \times \mathbb{R}^d \to \mathbb{R}^d\} \quad (1)$$

*where* $d < n$, $f$ *refers to a stack of standard DiT blocks whose width is smaller than the token dimension from the representation encoder, and* $\boldsymbol{A}, \boldsymbol{B}$ *denote the input and output linear projections, respectively. Then for **any** $g \in \mathcal{G}_d$,*

$$\mathcal{L}(g, \theta) = \int_0^1 \mathbb{E}_{\mathbf{x} \sim p(\mathbf{x}), \boldsymbol{\varepsilon} \sim \mathcal{N}(0, \mathbf{I}_n)} \big[\|g(\mathbf{x}_t, t) - (\boldsymbol{\varepsilon} - \mathbf{x})\|^2\big] \mathrm{d}t \geq \sum_{i=d+1}^n \lambda_i \quad (2)$$

*where* $\lambda_i$ *are the eigenvalues of the covariance matrix of the random variable* $W = \boldsymbol{\varepsilon} - \mathbf{x}$.

*Notably, when* $d \geq n$, $\mathcal{G}_d$ *contains the unique minimizer to* $\mathcal{L}(g, \theta)$.

*Proof.* See Appendix B.1. □

In our toy setting where $p(\mathbf{x}) = \delta(\mathbf{x} - \mathbf{x}_0)$, we have $W \sim \mathcal{N}(-\mathbf{x}, \mathbf{I}_n)$ and $\lambda_i = 1$ for all $i$. Thus by Theorem 1, the lower bound of the average loss becomes $\tilde{\mathcal{L}}(g, \theta) \geq \frac{1}{n}\sum_{i=d+1}^n 1 = \frac{n-d}{n}$. As shown in Figure 3, this theoretical bound is consistent with our empirical results.

We further extend our investigation to a more practical setting by examining three models of varying width, {DiT-S, DiT-B, DiT-L}. Each model is overfit on a single image encoded by {DINOv2-S, DINOv2-B, DINOv2-L}, respectively, corresponding to different token dimensions. As shown in Section 3.1, convergence occurs only when the model width is at least as large as the token dimension (e.g., DiT-B with DINOv2-B), while the loss fails to converge otherwise (e.g., DiT-S with DINOv2-B).

| | DiT-S | DiT-B | DiT-L |
|---|---|---|---|
| DINOv2-S | 3.6e−2 ✓ | 1.0e−3 ✓ | 9.7e−4 ✓ |
| DINOv2-B | 5.2e−1 ✗ | 2.4e−2 ✓ | 1.3e−3 ✓ |
| DINOv2-L | 6.5e−1 ✗ | 2.7e−1 ✗ | 2.2e−2 ✓ |

Table 3: **Overfitting losses.** Compared between different combinations of model width and token dimension.

> • **Suboptimal design for diffusion transformers.** *We now fix the width of DiT to be at least as large as the RAE token dimension. For RAE with the DINOv2-B encoder, we pair it with DiT-XL in our following experiments.*

## 3.2 DIMENSION-DEPENDENT NOISE SCHEDULE SHIFT

Many prior works (Teng et al., 2023; Chen, 2023; Hoogeboom et al., 2023; Esser et al., 2024) have observed that, for inputs $\mathbf{z} \in \mathbb{R}^{C \times H \times W}$, increasing the spatial resolution ($H \times W$) reduces information corruption at the same noise level, impairing diffusion training. These findings, however,

are based mainly on pixel- or VAE-encoded inputs with few channels (e.g., $C \leq 16$). In practice, the Gaussian noise is applied to both spatial and channel dimensions; as the number of channels increases, the effective "resolution" per token also grows, reducing information corruption further. We therefore argue that proposed resolution-dependent strategies in these prior works should be generalized to the *effective data dimension*, defined as the number of tokens times their dimensionality.

We adopt the shifting strategy of Esser et al. (2024): for a schedule $t_n \in [0,1]$ and input dimensions $n, m$, the shifted timestep is defined as $t_m = \frac{\alpha t_n}{1+(\alpha-1)t_n}$ where $\alpha = \sqrt{m/n}$ is a dimension-dependent scaling factor. We follow (Esser et al., 2024) in using $n = 4096$ as the base dimension and set $m$ to the effective data dimension of RAE. As shown in Table 4, this yields significant performance gains, showing its importance for training diffusion models in the high-dimensional RAE latent space.

|  | gFID |
|---|---|
| w/o shift | 23.08 |
| w/ shift | 4.81 |

Table 4: **Impact of schedule shift.**

> • **Suboptimal noise scheduling.** *We now default the noise schedule to be dependent on the effective data dimension for all our following experiments.*

### 3.3 NOISE-AUGMENTED DECODING

Unlike VAEs, where latent tokens are encoded as a continuous distribution $\mathcal{N}(\mu, \sigma^2 \mathbf{I})$ (Kingma & Welling, 2014), the RAE decoder $D$ is trained to reconstruct images from the discrete distribution $p(\mathbf{z}) = \sum_i \delta(\mathbf{x} - \mathbf{z}_i)$, where $\{\mathbf{z}_i\}$ denotes the training set processed by the RAE encoder $E$. At inference time, however, the diffusion model may generate latents that are noisy or deviate slightly from the training distribution due to imperfect training and sampling Abuduweili et al. (2024). This could introduce a notable out-of-distribution challenge for $D$, which degrades sampling quality.

To mitigate this issue, inspired by prior works on Normalizing Flows (Dinh et al., 2017; Ho et al., 2019; Zhai et al., 2025), we augment the RAE decoder training with an additive noise $\mathbf{n} \sim \mathcal{N}(0, \sigma^2 \mathbf{I})$. Concretely, rather than decoding directly from the clean latent distribution $p(\mathbf{z})$, we train $D$ on a smoothed distribution $p_{\mathbf{n}}(\mathbf{z}) = \int p(\mathbf{z} - \mathbf{n}) \mathcal{N}(0, \sigma^2 \mathbf{I})(\mathbf{n}) d\mathbf{n}$ to enhance the decoder's generalization to the denser output space of diffusion models. We further introduce stochasticity into $\sigma$ by sampling it from $|\mathcal{N}(0, \tau^2)|$, which helps regularize training and improve robustness.

|  | gFID | rFID |
|---|---|---|
| $\mathbf{z} \sim p(\mathbf{z})$ | 4.81 | 0.49 |
| $\mathbf{z} \sim p_{\mathbf{n}}(\mathbf{z})$ | 4.28 | 0.57 |

Table 5: **Impact of $p_{\mathbf{n}}(\mathbf{z})$.**

We analyze how $p_{\mathbf{n}}(\mathbf{z})$ affects reconstruction and generation. As shown in Table 5, it improves gFID but slightly worsens rFID. This trade-off is expected: adding noise smooths the latent distribution and, therefore, helps reduce OOD issues for the decoder, but also removes fine details, lowering reconstruction quality. We conduct more ablation experiments on $\tau$ and different encoders in Appendix G.2.

> • **Diffusion generates noisy latents.** *We now adopt the noise-augmented decoding for all our following experiments.*

We combine all of the above techniques to train a DiT-XL model on RAE latents. Our improved diffusion recipe achieves a gFID of **4.28** (Figure 4) after only 80 epochs and **2.39** after 720 epochs in RAE's latent space. With same model size, this not only surpasses prior diffusion baselines (e.g., SiT-XL (Ma et al., 2024)) trained on VAE latents (achieving a $47\times$ training speedup), but also outperforms the convergence speed of recent methods based on representation alignment (e.g., REPA-XL (Yu et al., 2025)), achieving a $16\times$ training speedup. In the following sections, we investigate ways to make RAE generation more efficient and effective, pushing it toward state-of-the-art performance.

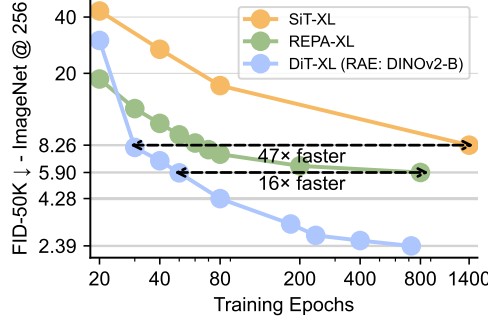

Figure 4: **DiT w/ RAE reaches much faster convergence and better FID than SiT or REPA.**

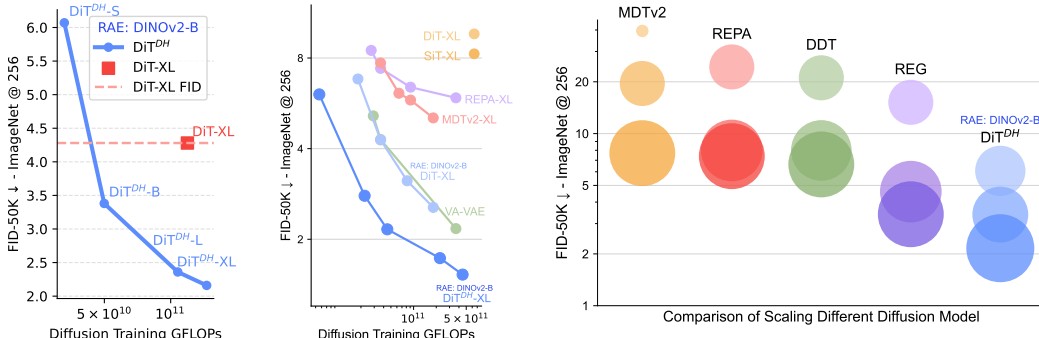

(a) DiT$^{DH}$ scales much better than DiT with RAE latents.

(b) DiT$^{DH}$ with RAE converges faster than VAE-based methods.

(c) DiT$^{DH}$ with RAE reaches better FID than VAE-based methods at all model scales. Bubble area indicates the flops of the model.

Figure 6: **Scalability of DiT$^{DH}$.** With RAE latents, DiT$^{DH}$ scales more efficiently in both training compute and model size than RAE-based DiT and VAE-based methods.

# 4 IMPROVING THE MODEL SCALABILITY WITH WIDE DIFFUSION HEAD

As discussed in Section 3, within the standard DiT framework, handling higher-dimensional RAE latents requires scaling up the width of the entire backbone, which quickly becomes computationally expensive. To overcome this limitation, we draw inspiration from DDT (Wang et al., 2025c) and introduce the DDT head—a *shallow yet wide* transformer module dedicated to denoising. By attaching this head to a standard DiT, we effectively increase model width without incurring quadratic growth in FLOPs. We refer to this augmented architecture as DiT$^{DH}$ throughout the remainder of the paper. We also conduct experiment of the design choice of DDT head in Appendix G.3

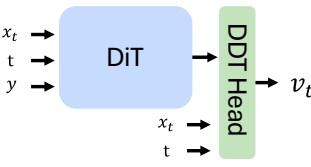

Figure 5: **The Wide DDT Head.**

**Wide DDT head.** Formally, a DiT$^{DH}$ model consists of a base DiT $M$ and an additional wide, shallow transformer head $H$. Given a noisy input $x_t$, timestep $t$, and an optional class label $y$, the combined model predicts the velocity $v_t$ as

$$z_t = M(x_t \mid t, y),$$
$$v_t = H(x_t \mid z_t, t),$$

**DiT$^{DH}$ converges faster than DiT.** We train a series of DiT$^{DH}$ models with varying backbone sizes (DiT$^{DH}$-S, B, L, and XL) on RAE latents. We use a 2-layer, 2048-dim DDT head for all DiT$^{DH}$ models. Performance is compared against the standard DiT-XL baseline. As shown in Figure 6a, DiT$^{DH}$ is substantially more FLOP-efficient than DiT. For example, DiT$^{DH}$-B requires only ~40% of the training FLOPs yet outperforms DiT-XL by a large margin; when scaled to DiT$^{DH}$-XL under a comparable training budget, DiT$^{DH}$ achieves an FID of 2.16—nearly half that of DiT-XL.

| Model | DINOv2 | | |
| --- | --- | --- | --- |
| | S | B | L |
| DiT-XL | 3.50 | 4.28 | 6.09 |
| DiT$^{DH}$-XL | 2.42 | 2.16 | 2.73 |

Table 6: **DiT$^{DH}$ outperforms DiT across RAE encoder sizes.**

**DiT$^{DH}$ maintains its advantage across RAE scales.** We compare DiT$^{DH}$-XL and DiT-XL on three RAE encoders—DINOv2-S, DINOv2-B, and DINOv2-L. As shown in Section 4, DiT$^{DH}$ consistently outperforms DiT, and the advantage grows with encoder size. For example, with DINOv2-L, DiT$^{DH}$ improves FID from 6.09 to 2.73. We attribute this robustness to the DDT head. Larger encoders produce higher-dimensional latents, which amplify the width bottleneck of DiT. DiT$^{DH}$ addresses this by satisfying the width requirement discussed in Section 3 while keeping features compact. It also filters out noisy information that becomes more prevalent in high-dimensional RAE latents.

## 4.1 STATE-OF-THE-ART DIFFUSION TRANSFORMERS

**Convergence.** We compare the convergence behavior of DiT$^{DH}$-XL with previous state-of-the-art diffusion models (Peebles & Xie, 2023; Ma et al., 2024; Yu et al., 2025; Gao et al., 2023; Yao

| Method | Epochs | #Params | Generation@256 w/o guidance | | | | Generation@256 w/ guidance | | | |
|---|---|---|---|---|---|---|---|---|---|---|
| | | | gFID↓ | IS↑ | Prec.↑ | Rec.↑ | gFID↓ | IS↑ | Prec.↑ | Rec.↑ |
| *Autoregressive* | | | | | | | | | | |
| VAR (Tian et al., 2024) | 350 | 2.0B | 1.92 | **323.1** | 0.82 | 0.59 | 1.73 | **350.2** | 0.82 | 0.60 |
| MAR (Li et al., 2024b) | 800 | 943M | 2.35 | 227.8 | 0.79 | 0.62 | 1.55 | 303.7 | 0.81 | 0.62 |
| xAR (Ren et al., 2025) | 800 | 1.1B | - | - | - | - | 1.24 | 301.6 | **0.83** | 0.64 |
| *Pixel Diffusion* | | | | | | | | | | |
| ADM (Dhariwal & Nichol, 2021) | 400 | 554M | 10.94 | 101.0 | 0.69 | 0.63 | 3.94 | 215.8 | **0.83** | 0.53 |
| RIN (Jabri et al., 2023) | 480 | 410M | 3.42 | 182.0 | - | - | - | - | - | - |
| PixelFlow (Chen et al., 2025e) | 320 | 677M | - | - | - | - | 1.98 | 282.1 | 0.81 | 0.60 |
| PixNerd (Wang et al., 2025b) | 160 | 700M | - | - | - | - | 2.15 | 297.0 | 0.79 | 0.59 |
| SiD2 (Hoogeboom et al., 2025) | 1280 | - | - | - | - | - | 1.38 | - | - | - |
| *Latent Diffusion with VAE* | | | | | | | | | | |
| DiT (Peebles & Xie, 2023) | 1400 | 675M | 9.62 | 121.5 | 0.67 | 0.67 | 2.27 | 278.2 | **0.83** | 0.57 |
| MaskDiT (Zheng et al.) | 1600 | 675M | 5.69 | 177.9 | 0.74 | 0.60 | 2.28 | 276.6 | 0.80 | 0.61 |
| SiT (Ma et al., 2024) | 1400 | 675M | 8.61 | 131.7 | 0.68 | 0.67 | 2.06 | 270.3 | 0.82 | 0.59 |
| MDTv2 (Gao et al., 2023) | 1080 | 675M | - | - | - | - | 1.58 | 314.7 | 0.79 | 0.65 |
| VA-VAE (Yao et al., 2025) | 80 | 675M | 4.29 | - | - | - | - | - | - | - |
| | 800 | | 2.17 | 205.6 | 0.77 | 0.65 | 1.35 | 295.3 | 0.79 | 0.65 |
| REPA (Yu et al., 2025) | 80 | 675M | 7.90 | 122.6 | 0.70 | 0.65 | - | - | - | - |
| | 800 | | 5.78 | 158.3 | 0.70 | 0.68 | 1.29 | 306.3 | 0.79 | 0.64 |
| DDT (Wang et al., 2025c) | 80 | 675M | 6.62 | 135.2 | 0.69 | 0.67 | 1.52 | 263.7 | 0.78 | 0.63 |
| | 400 | | 6.27 | 154.7 | 0.68 | **0.69** | 1.26 | 310.6 | 0.79 | 0.65 |
| REPA-E (Leng et al., 2025) | 80 | 675M | 3.46 | 159.8 | 0.77 | 0.63 | 1.67 | 266.3 | 0.80 | 0.63 |
| | 800 | | 1.70 | 217.3 | 0.77 | 0.66 | 1.15 | 304.0 | 0.79 | 0.66 |
| *Latent Diffusion with RAE (Ours)* | | | | | | | | | | |
| DiT-XL (DINOv2-S) | 800 | 676M | 1.87 | 209.7 | 0.80 | 0.63 | 1.41 | 309.4 | 0.80 | 0.63 |
| DiT$^{DH}$-XL (DINOv2-B) | 20 | 839M | 3.71 | 198.7 | **0.86** | 0.50 | – | – | – | – |
| | 80 | | 2.16 | 214.8 | 0.82 | 0.59 | – | – | – | – |
| | 800 | | **1.51** | 242.9 | 0.79 | 0.63 | **1.13** | 262.6 | 0.78 | **0.67** |

Table 8: **Class-conditional performance on ImageNet 256×256.** RAE reaches an FID of 1.51 without guidance, outperforming all prior methods by a large margin. It also achieves an FID of 1.13 with AutoGuidance (Karras et al., 2025). We identified an inconsistency in the FID evaluation protocol in prior literature and re-ran the sampling process for several baselines. This resulted in higher baseline numbers than those originally reported. Further details are discussed in Section 4.1.

et al., 2025) in terms of FID without guidance. In Figure 6b, we show the convergence curve of DiT$^{DH}$-XL with training epochs/GFLOPs, while baseline models are plotted at their reported final performance. DiT$^{DH}$-XL already surpasses REPA-XL, MDTv2-XL, and SiT-XL around $5 \times 10^{10}$ GFLOPs, and by $5 \times 10^{11}$ GFLOPs it achieves the best FID overall, requiring over $40 \times$ less compute.

**Scaling.** We compare DiT$^{DH}$ with recent methods of different model at different scales. As shown in Figure 6c, increasing the size of DiT$^{DH}$ consistently improves the FID scores. The smallest model, DiT$^{DH}$-S, reaches a competitive FID of 6.07, already outperforming the much larger REPA-XL. When scaling from DiT$^{DH}$-S to DiT$^{DH}$-B, the FID improves significantly from 6.07 to 3.38, surpassing all prior works of similar or even larger scale. The performance continues to improve with DiT$^{DH}$-XL, setting a new state-of-the-art result of 2.16 at 80 training epochs.

**Performance.** Finally, we provide a quantitative comparison between DiT$^{DH}$-XL, our most performant model, with recent state-of-the-art diffusion models on ImageNet $256 \times 256$ and $512 \times 512$ in Table 8 and Table 7. Our method outperforms all prior diffusion models by a large margin, setting new state-of-the-art FID scores of **1.51** with-

| Method | Generation@512 | | | |
|---|---|---|---|---|
| | gFID↓ | IS↑ | Prec.↑ | Rec.↑ |
| BigGAN-deep (Brock et al., 2019) | 8.43 | 177.9 | **0.88** | 0.29 |
| StyleGAN-XL (Sauer et al., 2022) | 2.41 | 267.8 | 0.77 | 0.52 |
| VAR (Tian et al., 2024) | 2.63 | 303.2 | - | - |
| MAGVIT-v2 (Yu et al., 2024a) | 1.91 | **324.3** | - | - |
| XAR (Ren et al., 2025) | 1.70 | 281.5 | - | - |
| ADM | 3.85 | 221.7 | 0.84 | 0.53 |
| SiD2 | 1.50 | - | - | - |
| DiT | 3.04 | 240.8 | 0.84 | 0.54 |
| SiT | 2.62 | 252.2 | 0.84 | 0.57 |
| DiffiT (Hatamizadeh et al., 2024) | 2.67 | 252.1 | 0.83 | 0.55 |
| REPA | 2.08 | 274.6 | 0.83 | 0.58 |
| DDT | 1.28 | 305.1 | 0.80 | **0.63** |
| EDM2 (Karras et al., 2024) | 1.25 | - | - | - |
| DiT$^{DH}$-XL (DINOv2-B) | **1.13** | 259.6 | 0.80 | **0.63** |

Table 7: **Class-conditional performance on ImageNet 512×512 with guidance.** DiT$^{DH}$ with 400-epoch training achieves an strong FID score of 1.13.

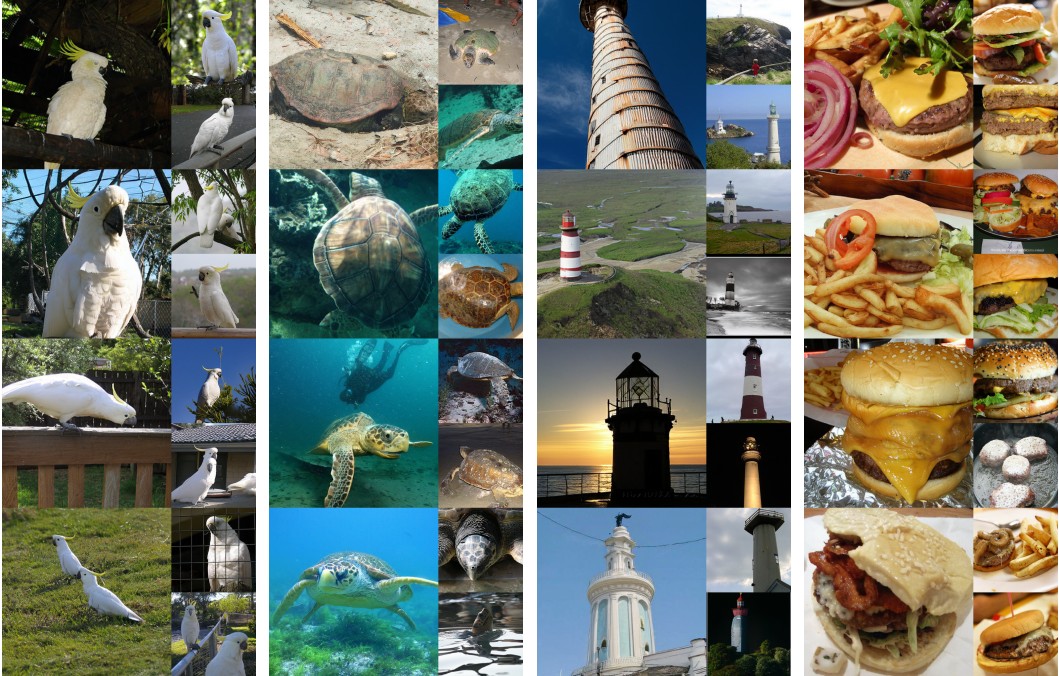

Figure 7: **Qualitative samples** from our model trained at $512 \times 512$ resolution with AutoGuidance. The RAE-based DiT demonstrates strong diversity, fine-grained detail, and high visual quality.

out guidance and **1.13** with guidance at $256 \times 256$. On $512 \times 512$, with 400-epoch training, DiT$^{\text{DH}}$-XL further achieves an FID of **1.13** with guidance, surpassing the previous best performance achieved by EDM-2 (1.25).

Following the quantitative results, we present qualitative samples from our best models in Figure 7. These visualizations exhibit both high semantic diversity and fine-grained details comparable to ground-truth ImageNet samples, consistent with the achieved state-of-the-art FID. We provide additional visualization samples in Appendix N and unconditional generation results in Appendix L.

**Remarks on FID evaluation.** To construct the 50,000 samples used for conditional FID evaluation, we note that previous works such as DDT (Wang et al., 2025c), VAR (Tian et al., 2024), MAR (Li et al., 2024b), and xAR (Ren et al., 2025) typically evaluate using exactly 50 images per class, while others employ uniform random sampling across the 1,000 class labels. Although uniform random sampling asymptotically approaches the balanced sampling case, we surprisingly observed that class-balanced sampling consistently achieves around 0.1 lower FID scores. To ensure fair comparison, we re-evaluate several recent methods with accessible checkpoints, such as SiT (Peebles & Xie, 2023), REPA (Yu et al., 2025), REPA-E (Leng et al., 2025) using class-balanced sampling and update their reported scores accordingly. A more detailed comparison is provided in Appendix E.

## 5 DISCUSSIONS

### 5.1 HOW CAN RAE EXTEND TO HIGH-RESOLUTION SYNTHESIS EFFICIENTLY?

A central challenge in generating high-resolution images is that resolution scales with the number of tokens: doubling image size in each dimension requires roughly four times as many tokens. To address this, we let the decoder handle resolution scaling by allowing its patch size patch size $p_d$ to differ from the encoder patch size $p_e$. When $p_d = p_e$, the output matches the input resolution; setting $p_d = 2p_e$ produces a $2\times$ upsampled image, reconstructing a $512 \times 512$ image from the same tokens used at $256 \times 256$.

Since the decoder is decoupled from both the encoder and the diffusion process, we can reuse diffusion models trained at $256 \times 256$ resolution, simply swapping in an upsampling decoder to produce $512 \times 512$ outputs without retraining. As shown in Table 9, this approach slightly increases rFID but achieves competitive gFID, while being $4\times$ more efficient than quadrupling the number of tokens.

| Method | #Tokens | gFID ↓ | rFID ↓ |
|---|---|---|---|
| Direct | 1024 | 1.13 | 0.53 |
| Upsample | 256 | 1.61 | 0.97 |

Table 9: **Comparison on ImageNet $512 \times 512$.** Decoder upsampling achieves competitive FID compared to direct 512-resolution training. Both models are trained for 400 epochs.

## 5.2 DOES DiT^DH WORK WITHOUT RAE?

In this work, we propose and study RAE and DiT^DH. In Section 3, we showed that RAE with DiT already brings substantial benefits, even in the absence of DiT^DH. Here, we turn the question around: can DiT^DH still provide improvements, without the latent space of RAE?

|  | VAE | DINOv2-B |
|---|---|---|
| DiT-XL | 7.13 | 4.28 |
| DiT^DH-XL | 11.70 | 2.16 |

Table 10: **Performance on VAE.** DiT^DH yields worse FID than DiT, despite using extra compute for the wide DDT head.

To investigate, we train both DiT-XL and DiT^DH-XL on SD-VAE latents with a patch size of 2, alongside DINOv2-B for comparison, for 80 epochs, and report unguided FID. As shown in Table 10, DiT^DH-XL performs even worse than DiT-XL on SD-VAE, despite the additional computation introduced by the diffusion head. This indicates that the DDT head provides little benefit in low-dimensional latent spaces, and its primary strength arises in high-dimensional diffusion tasks introduced by RAE.

## 5.3 THE ROLE OF STRUCTURED REPRESENTATION IN HIGH-DIMENSIONAL DIFFUSION?

DiT^DH achieves strong performance when paired with the high-dimensional latent space of RAE. This raises a key question: is the structured representation of RAE essential, or would DiT^DH work equally well on unstructured high-dimensional inputs such as *raw pixels*?

To evaluate this, we train DiT-XL and DiT^DH-XL directly on raw pixels. For $256 \times 256$ images with a patch size of 16, the resulting DiT input token dimensionality is $16 \times 16 \times 3 = 768$, matching that of the DINOv2-B latents. We report unguided FID after 80 epochs. As shown in Table 11, DiT^DH outperforms DiT on pixels, but both models perform far worse than their counterparts trained on RAE latents. These results demonstrate that high dimensionality alone is not sufficient: the structured representation provided by RAE is crucial for achieving strong performance gains.

|  | Pixel | DINOv2-B |
|---|---|---|
| DiT-XL | 51.09 | 4.28 |
| DiT^DH-XL | 30.56 | 2.16 |

Table 11: **Comparison on pixel diffusion.** Pixel Diffusion has much worse FID than diffusion on DINOv2-B.

## 6 RELATED WORKS

Here, we discuss previous work on the line of representation learning and reconstruction/generation. We present a more detailed related work discussion in Appendix A.

**Representation for Generation and Reconstruction.** Recent work enhances generative modeling with semantic representations for both reconstruction and generation. For reconstruction, VA-VAE (Yao et al., 2025) aligns VAE latents with a pretrained representation encoder, while MAETok (Chen et al., 2025a), DC-AE 1.5 (Chen et al., 2025d), and l-DEtok (Yang et al., 2025) incorporate MAE- or DAE (Vincent et al., 2008)-inspired objectives into VAE training, substantially improving reconstruction and generation quality, yet still relying on heavily compressed, low-dimensional latents. For generation, REPA (Yu et al., 2025) aligns intermediate DiT features with representation encoders representations, DDT (Wang et al., 2025c) decouples DiT into an encoder–decoder with REPA-style supervision, REG (Wu et al., 2025) introduces a learnable token aligned to a representation encoders representation, and ReDi (Kouzelis et al., 2025b) jointly generates VAE latents and PCA components of DINOv2 features within diffusion models. In contrast, we eliminate latent compression and auxiliary alignment: we reconstruct directly from representation encoders features using a simple ViT decoder and train diffusion models directly in the representation encoders space, achieving reconstruction quality comparable to or better than SD-VAE (Rombach et al., 2022) while preserving stronger representations and enabling faster convergence.

## 7 ACKNOWLEGMENTS

The authors would like to thank Sihyun Yu, Jaskirat Singh, Shusheng Yang, Xichen Pan, Chenyu Li, Bingda Tang, Fred Lu, David Fan, Amir Bar, Shangyuan Tong for insightful discussions and feedback on the manuscript. This work was mainly supported by the Google TPU Research Cloud (TRC) program and the Opend Path AI Foundation. ST is supported by Meta AI Mentorship Program. SX also acknowledges support from the MSIT IITP grant (RS-2024-00457882) and the NSF award IIS-2443404.

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

## A    EXTENDED RELATED WORK

**Representation encoder as autoencoder.**    Recent studies have investigated leveraging semantic representations for reconstruction, particularly in MLLMs where diffusion decoders are conditioned on semantic tokens (Sun et al., 2024; Chen et al., 2025b; Pan et al., 2025; Tong et al., 2025). While this improves visual quality, the reliance on large pretrained diffusion decoders makes reconstructions less faithful to the input, limiting their effectiveness as true autoencoders. Very recently, UniLIP (Tang et al., 2025) employs a one-step convolutional decoder on top of InternViT (Zhu et al., 2025), achieving reconstruction quality surpassing SD-VAE. However, UniLIP relies on additional large-scale fine-tuning of pretrained ViTs, arguing that frozen pretrained representation encoders lacks sufficient visual detail. In contrast, we show this is not the case: frozen representation encoders achieves comparable reconstruction performance while enabling much faster convergence in diffusion training.

Another line of related work also try to utilize representation encoders directly as tokenizers. VFM-Tok (Zheng et al., 2025) and DiGIT (Zhu et al., 2024b) applies vector-quantization directly to pretrained representation encoders like Dino or SigLIP. These approaches transform representation encoders into an effective tokenizer for AR models, but still suffer from the information capacity bottleneck brought by quantization.

**Compressed image tokenizers.**    Autoencoders have long been used to compress images into low-dimensional representations for reconstruction (Hinton & Salakhutdinov, 2006; Vincent et al., 2008). VAEs (Kingma & Welling, 2014) extend this paradigm by mapping inputs to Gaussian distributions, while VQ-VAEs (Oord et al., 2017; Razavi et al., 2019) introduce discrete latent codes. VQGAN (Esser et al., 2021) adds adversarial objectives, and ViT-VQGAN (Esser et al., 2021; Cao et al., 2023) modernizes the architecture with Vision Transformers (ViTs) (Dosovitskiy et al., 2021). Other advances include multi-stage quantization (Lee et al., 2022; Zheng et al., 2022), lookup-free schemes (Mentzer et al., 2024; Zhao et al., 2025), token-efficient designs such as TiTok and DCAE (Yu et al., 2024b; Chen et al., 2025c), and structure-preserving approaches like EQ-VAE (Kouzelis et al., 2025a). (Hansen-Estruch et al., 2025) further explores the scaling behavior of VAEs. LARP, CRT, REPA-E (Wang et al., 2025a; Ramanujan et al., 2025; Yu et al., 2025) tried to improve VAE with generative priors via back-propagation. In contrast, we dispense with aggressive compression and instead adopt a pretrained representation encoders as encoder. This avoids the encoder collapsing into shallow features optimized only for reconstruction loss, while providing strong pretrained representations that serve as a robust latent space.

**Generative models.**    Modern image generation is dominated by two paradigms: autoregressive (AR) models and diffusion models. AR models (Ramesh et al., 2022; Yu et al., 2022; Parmar et al., 2018; Chen et al., 2020a) generate images sequentially, token by token, and benefit from powerful language-model architectures but often suffer from slow sampling. Diffusion models (Zhu et al., 2024a; Ho et al., 2020; Nichol & Dhariwal, 2021; Rombach et al., 2022; Ma et al., 2024; Peebles & Xie, 2023) instead learn to iteratively denoise noisy signals, offering superior sample quality and scalability, though at the cost of many sampling steps. In this work, we build on diffusion models but adapt them to high-dimensional latent spaces provided by pretrained representation encoders. We find representation encoders provide faster convergence and improved scaling behavior.

**Robust decoders for generation.**    Recent works suggest that incorporating masking or latent denoising losses can improve tokenizer training. *l*-DeTok (Yang et al., 2025) shows that combining both losses yields strong VAEs for second-stage MAR (Li et al., 2024b; Fan et al., 2025b) generation, while RobusTok (Qiu et al., 2025) demonstrates that training with perturbed tokens makes decoders more robust. Notably, Yang et al. (2025) report that denoising loss only increase performance when encoder and decoder are jointly trained, whereas we observe substantial gains in generation quality with frozen encoders.

**Visual representation learning.**    Visual representations primarily fall into two broad families: self-supervised encoders that learn invariances from augmented views or masked prediction (He et al., 2019; Chen et al., 2020b; Grill et al., 2020; Zhou et al., 2021; He et al., 2021), and language-supervised encoders trained from image–text pairs (Radford et al., 2021; Zhai et al., 2023; Chen et al., 2024; Tschannen et al., 2025). Recent work (Fan et al., 2025a; Tschannen et al., 2025; Wang

et al., 2025d) scales both lines on billion-scale corpora, yielding robust, semantically rich features. In this work, we leverage these pretrained encoders directly: we freeze the encoder, and train a lightweight decoder to form a Representation Autoencoder (RAE). We show that such RAEs are effective for both reconstruction and generation.

# B    PROOFS

## B.1    PROOF OF LOWER BOUND FOR TRAINING LOSS

**Theorem 1.** *Assuming* $\mathbf{x} \sim p(\mathbf{x}) \in \mathbb{R}^n, \boldsymbol{\varepsilon} \sim \mathcal{N}(0, \mathbf{I}_n), t \in [0, 1]$. *Let* $\mathbf{x}_t = (1 - t)\mathbf{x} + t\boldsymbol{\varepsilon}$, *consider the function family*

$$\mathcal{G}_d = \{g(\mathbf{x}_t, t) = \boldsymbol{B}f(\boldsymbol{A}\mathbf{x}_t, t) : \boldsymbol{A} \in \mathbb{R}^{d \times n}, \boldsymbol{B} \in \mathbb{R}^{n \times d}, f : [0, 1] \times \mathbb{R}^d \to \mathbb{R}^d\} \tag{1}$$

*where* $d < n$, $f$ *refers to a stack of standard DiT blocks whose width is smaller than the token dimension from the representation encoder, and* $\boldsymbol{A}, \boldsymbol{B}$ *denote the input and output linear projections, respectively. Then for **any** $g \in \mathcal{G}_d$,*

$$\mathcal{L}(g, \theta) = \int_0^1 \mathbb{E}_{\mathbf{x} \sim p(\mathbf{x}), \boldsymbol{\varepsilon} \sim \mathcal{N}(0, \mathbf{I}_n)} \left[ \|g(\mathbf{x}_t, t) - (\boldsymbol{\varepsilon} - \mathbf{x})\|^2 \right] \mathrm{d}t \geq \sum_{i=d+1}^n \lambda_i \tag{2}$$

*where* $\lambda_i$ *are the eigenvalues of the covariance matrix of the random variable* $W = \boldsymbol{\varepsilon} - \mathbf{x}$.

*Notably, when* $d \geq n$, $\mathcal{G}_d$ *contains the unique minimizer to* $\mathcal{L}(g, \theta)$.

*Proof.* By Albergo et al. (2023), the distribution $\rho_t$ of $\mathbf{x}_t$ satisfies $\rho_0 = p(\mathbf{x}), \rho_1 = \mathcal{N}(0, \mathbf{I}_n)$, and $\partial_t \rho + \nabla \cdot (v\rho) = 0$ where $v$ is the optimal velocity predictor defined as $v(\mathbf{x}_t, t) = \mathbb{E}[\boldsymbol{\varepsilon} - \mathbf{x} | \mathbf{x}_t]$. Also, by Theorem 2.7 in Albergo et al. (2023), there exists $f^* \in C^0((C^1(\mathbb{R}^n))^n; [0, 1])$[1] that uniquely minimizes the $\mathcal{L}(f, \theta)$ and perfectly approximates $v$.

By our training setting, it's reasonable to assume that $\mathbf{x} \sim p(\mathbf{x})$ and $\boldsymbol{\varepsilon} \sim \mathcal{N}(0, \mathbf{I}_n)$ are independent. Then the distribution of the objective $\mathbf{y} = \boldsymbol{\varepsilon} - \mathbf{x} \sim p_{\mathbf{y}}(\mathbf{y})$ satisfies $p_{\mathbf{y}}(\mathbf{y}) = \int_{\mathbb{R}^n} \mathcal{N}(0, \mathbf{I}_n)(\mathbf{y} + \mathbf{x})p(\mathbf{x})\mathrm{d}\mathbf{x}$. Clearly, $p_{\mathbf{y}}$ has full support on $\mathbb{R}^n$ and is strictly positive, indicating $\mathbf{y}$ has a non-zero probability anywhere in $\mathbb{R}^n$. Similarly, for $\mathbf{x}_t = (1 - t)\mathbf{x} + t\boldsymbol{\varepsilon}$, given any $t$, $p_{\mathbf{x}_t}(\mathbf{w}) = \int_{\mathbb{R}^n} \mathcal{N}(0, t^2 \mathbf{I}_n)(\mathbf{w} - \mathbf{x})\frac{1}{(1-t)}p(\frac{\mathbf{x}}{1-t})\mathrm{d}\mathbf{x}$ also has full support on $\mathbb{R}^n$ and is strictly positive, indicating $\mathbf{x}_t$ has a non-zero probability anywhere in $\mathbb{R}^n$ as well.

Recall that for any function $f : \mathcal{X} \to \mathcal{Y}$, $\mathrm{Im}(f) = \{f(x) : x \in \mathcal{X}\}$. Then for linear transformation $f(\mathbf{x}) = \boldsymbol{M}\mathbf{x}$ with $\boldsymbol{M} \in \mathbb{R}^{d \times n}$, $\mathrm{Im}(f) = \{\boldsymbol{M}\mathbf{x} : \mathbf{x} \in \mathbb{R}^n\}$; we denote this as $\mathrm{Im}(\boldsymbol{M})$. Now, for any $g \in \mathcal{G}_d$, $\mathrm{Im}(g) = \{\boldsymbol{B}f(\boldsymbol{A}\mathbf{x}) : \mathbf{x} \in \mathbb{R}^n\} \subseteq \{\boldsymbol{B}\mathbf{y} : \mathbf{y} \in \mathbb{R}^d\} = \mathrm{Im}(\boldsymbol{B})$. Since $\mathrm{rank}(\boldsymbol{B}) \leq d$, $\dim \mathrm{Im}(g) \leq \mathrm{rank}(\boldsymbol{B}) \leq d < n$, therefore $\mathrm{Im}(g) \subseteq \mathrm{Im}(\boldsymbol{B}) \subset \mathbb{R}^n$.

Now, given $g \in \mathcal{G}_d$ and the deterministic pair $(\mathbf{x}_t, \mathbf{y}, t) \in (\mathbb{R}^n, \mathbb{R}^n, [0, 1])$, by Projection Theorem,

$$\|g(\mathbf{x}_t, t) - \mathbf{y}\|^2 \geq \|\mathbf{u}_g - \mathbf{y}\|^2 \tag{3}$$

where $\mathbf{u}_g \in \mathrm{Im}(g)$ is the unique minimizer and $\mathbf{u}_g - \mathbf{y}$ is orthogonal to $\mathrm{Im}(g)$. Since $\|\cdot\|^2 \geq 0$, we can take expectation on both sides

$$\inf_{g \in \mathcal{G}_d} \mathbb{E}_{\mathbf{x} \sim p(\mathbf{x}), \boldsymbol{\varepsilon} \sim \mathcal{N}(0, \mathbf{I}_n)} \left[ \|g(\mathbf{x}_t, t) - \mathbf{y}\|^2 \right] \geq \inf_{g \in \mathcal{G}_d} \mathbb{E} \left[ \|\mathbf{u}_g - \mathbf{y}\|^2 \right]$$

$$\geq \inf_{\mathbf{u} \in S; \dim S \leq d} \mathbb{E} \left[ \|\mathbf{u} - \mathbf{y}\|^2 \right]$$

$$\geq \inf_{S; \dim S \leq d} \mathbb{E} \left[ \|\mathbf{y}\|^2 - \|\boldsymbol{P}_S \mathbf{y}\|^2 \right] \tag{4}$$

---

[1]family of functions $f : [0, 1] \times \mathbb{R}^n \to \mathbb{R}^n$ that is continuous in $t$ for all $(\mathbf{x}, t) \in [0, 1] \times \mathbb{R}^n$, and $f(\cdot, t)$ is a continuously differentiable function from $\mathbb{R}^n$ to $\mathbb{R}^n$.

where $\boldsymbol{P}_S$ denote the projection matrix from $\mathbb{R}^n$ onto $S$. Without loss of generality, we assume $\mathbb{E}[\mathbf{x}] = 0^2$, then Eq 4 can be expanded as

$$
\begin{aligned}
\inf_{g \in \mathcal{G}_d} \mathbb{E}_{\mathbf{x} \sim p(\mathbf{x}), \boldsymbol{\varepsilon} \sim \mathcal{N}(0, \mathbf{I}_n)} \left[ \| g(\mathbf{x}_t, t) - \mathbf{y} \|^2 \right] &\geq \sum_{i=1}^{n} \mathbb{E}[\mathbf{y}_i^2] - \sup_{S; \dim S \leq d} \sum_{i=1}^{n} \mathbb{E}[(\boldsymbol{P}_S \mathbf{y})_i^2] \\
&= \mathrm{Tr}(\mathrm{Cov}(\mathbf{y})) - \sup_{S; \dim S \leq d} \mathrm{Tr}(\mathrm{Cov}(\boldsymbol{P}_S \mathbf{y})) \\
&\geq \sum_{i=1}^{n} \lambda_i - \sum_{i=1}^{d} \lambda_i = \sum_{i=d+1}^{n} \lambda_i
\end{aligned}
\tag{5}
$$

where Eq 5 is obtained via Ky-Fan Maximum Principle.

When $d \geq n$, $\sup_S \mathbb{E}[\|\boldsymbol{P}_S \mathbf{y}\|^2] = \mathbb{E}[\|\mathbf{y}\|^2]$, leading to a trivial lower bound in Eq 5, and $\mathcal{G}_d = C^0((C^1(\mathbb{R}^n))^n; [0,1])$. $\qquad \square$

### B.2 PROOF OF LOWER BOUND FOR INFERENCE LOSS

**Theorem 2.** *Consider the same setup as Theorem 1. Let $\mathbf{x}_1$ be the initial random variables in the sampling process, and*

$$
\begin{aligned}
\mathbf{x}_0 &= ODE(g, \mathbf{x}_1, 1 \rightarrow 0) \\
\mathbf{x}_0^* &= ODE(f^*, \mathbf{x}_1, 1 \rightarrow 0)
\end{aligned}
$$

*where $ODE(f, \mathbf{x}, t \rightarrow s)$ refers to any ODE solver that integrates $f$ from time $t$ to $s$ using $\mathbf{x}$ as the initial condition. We further assume for any $(\mathbf{x}, \mathbf{y}, t) \in (\mathbb{R}^n, \mathbb{R}^n, [0, 1])$, there exists constant $L > 0$ such that*

$$
\| f^*(\mathbf{x}, t) - f^*(\mathbf{y}, t) \| \leq L \| \mathbf{x} - \mathbf{y} \|
$$

*then*

$$
\| \mathbf{x}_0^* - \mathbf{x}_0 \| \geq \frac{1 - e^{-L}}{L} \sum_{i=d+1}^{n} \lambda_i
\tag{6}
$$

*Proof.* We first define a forward $ODE$ that integrates from $0 \rightarrow 1$ $\mathbf{x}_{\overleftarrow{t}} := \mathbf{x}_{-t}$, and

$$
\begin{aligned}
\mathrm{d}\mathbf{x}_{\overleftarrow{t}} &= g(\mathbf{x}_{\overleftarrow{t}}, t) \mathrm{d}t \\
\mathrm{d}\mathbf{x}_{\overleftarrow{t}}^* &= f^*(\mathbf{x}_{\overleftarrow{t}}^*, t) \mathrm{d}t
\end{aligned}
$$

Then

$$
\begin{aligned}
\frac{\mathrm{d}}{\mathrm{d}t} \| \mathbf{x}_{\overleftarrow{t}}^* - \mathbf{x}_{\overleftarrow{t}} \| &= \| f^*(\mathbf{x}_{\overleftarrow{t}}^*, t) - g(\mathbf{x}_{\overleftarrow{t}}, t) \| \\
&\geq \| f^*(\mathbf{x}_{\overleftarrow{t}}, t) - g(\mathbf{x}_{\overleftarrow{t}}, t) \| - \| f^*(\mathbf{x}_{\overleftarrow{t}}^*, t) - f^*(\mathbf{x}_{\overleftarrow{t}}, t) \| \\
&\geq \| \Delta \| - L \| \mathbf{x}_{\overleftarrow{t}}^* - \mathbf{x}_{\overleftarrow{t}} \|
\end{aligned}
\tag{7}
$$

where $\Delta$ denotes the approximation error to $f^*$ for $g \in \mathcal{G}_d$. Applying Gronwall's Lemma, we have

$$
e^{L \overleftarrow{t}} \| \mathbf{x}_{\overleftarrow{t}}^* - \mathbf{x}_{\overleftarrow{t}} \| \Big|_0^1 \geq \int_0^t e^{Ls} \| \Delta \| \mathrm{d}s
$$

$$
\implies \| \mathbf{x}_0^* - \mathbf{x}_0 \| = \| \mathbf{x}_{\overleftarrow{1}}^* - \mathbf{x}_{\overleftarrow{1}} \| \geq \frac{(1 - e^{-L})}{L} \sum_{i=d+1}^{n} \lambda_i
\tag{8}
$$

where by Theorem 1, $\| \Delta \| \geq \sum_{i=d+1}^{n} \lambda_i$. $\qquad \square$

---

$^2$Non-centered $\mathbf{x}$ with $\mathbb{E}[\mathbf{x}] = \mu$ will additionally introduce $\| \mu \|^2 - \| \boldsymbol{P}_S \mu \|^2$ to the lower bound; we ignore this term since most data processing pipelines will center the data.

## C RAE IMPLEMENTATION

### C.1 ENCODER NORMALIZATION

For any given frozen representation encoders, we discard any [CLS] or [REG] token produced by the encoder, and keep the all of patch tokens. We then apply a layer normalization to each token independently, to ensure each token has zero mean and unit variance across channels. We note that all representation encoders we use adopt the standard ViT architecture (Dosovitskiy et al., 2021), which have already applied layer normalization after the last transformer block. Therefore, we only need to cancel the affine parameters of the layer normalization in representation encoders. This does not affect the representation quality of representation encoders, as it is a linear transformation.

**Practical Notes.** Specifically, we use DINOv2 with Registers (Darcet et al., 2025). Since DINOv2 only provides variants with $p_e = 14$, we interpolate the input images to $224 \times 224$ but set $p_d = 16$, ensuring the model still produces 256 tokens while reconstructing $256 \times 256$ images.

### C.2 DECODER TRAINING DETAILS

**Datasets.** We primarily use ImageNet-1K for training all decoders. Most experiments are conducted at a resolution of $256 \times 256$. For 512-resolution synthesis without decoder upsampling, we train decoders directly on $512 \times 512$ images.

**Decoder Architecture.** The decoder takes the token embeddings produced by the frozen encoder takes the token embedding reconstructs them back into the pixel space using the same patch size as the encoder. As a result, it can generate images with the same spatial spatial resolution as the encoder's inputs input. Following He et al. (2021), we prepend a learnable [CLS] token decoder's input sequence and discard it after decoding.

**Discriminator Architecture.** We include the majority of our decoder training details in Table 12. We follow most of the design choices in StyleGAN-T (Sauer et al., 2023), except for using a frozen Dino-S/8 (Caron et al., 2021) instead of Dino-S/16 as the discriminator. We found using Dino-S/8 stabilizes training and avoid the decoder to generate adversarial patches. We also remove the virtual batch norm in Sauer et al. (2023) and use the standard batch norm instead. All input is interpolated to $224 \times 224$ resolution before feeding into the discriminator.

Table 12: Training configuration for decoder and discriminator.

| Component | Decoder | Discriminator |
|---|---|---|
| optimizer | Adam | Adam |
| max learning rate | $2 \times 10^{-4}$ | $2 \times 10^{-4}$ |
| min learning rate | $2 \times 10^{-5}$ | $2 \times 10^{-5}$ |
| learning rate schedule | cosine decay | cosine decay |
| optimizer betas | (0.5, 0.9) | (0.5, 0.9) |
| weight decay | 0.0 | 0.0 |
| batch size | 512 | 512 |
| warmup | 1 epoch | 1 epoch |
| loss | $\ell_1$ + LPIPS + GAN | adv. |
| Model | ViT-(B, L, XL) | Dino-S/8 (frozen) |
| LPIPS start epoch | 0 | – |
| disc. start epoch | – | 6 |
| adv. loss start epoch | 8 | – |
| Training epochs | 16 | 10 |

**Losses.** For training the decoder, we use a mixture of L1, LPIPS (Zhang et al., 2018) and adversarial loss (Goodfellow et al., 2014):

$$z = E(x), \hat{x} = D(z)$$
$$\mathcal{L}_{rec}(x) = \omega_L \, \text{LPIPS}(\hat{x}, x) + \text{L1}(\hat{x}, x) + \omega_G \lambda \, \text{GAN}(\hat{x}, x),$$

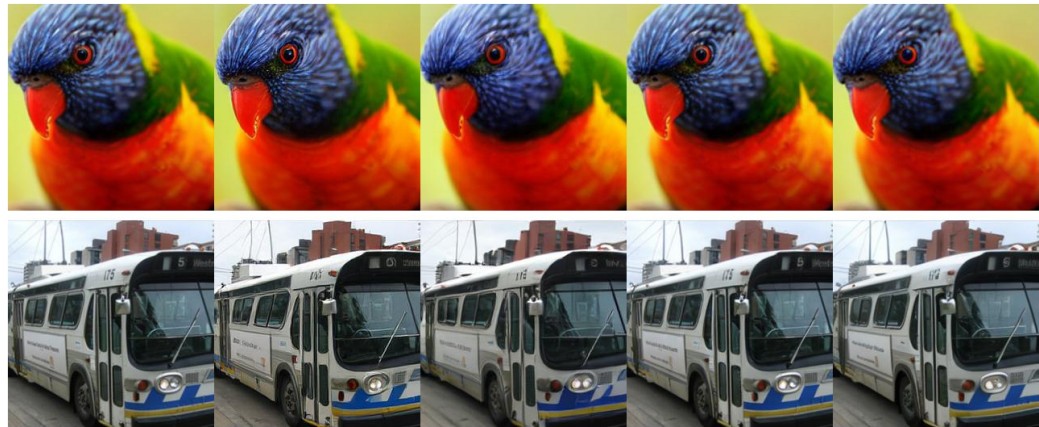

Figure 8: **Reconstruction examples.** From left to right: input image, RAE (DINOv2-B), RAE (SigLIP2-B), RAE (MAE-B), SD-VAE. Zoom in for details.

where $E, D$ are the encoder and decoder of RAE. We set $\omega_L = 1.$ and $\omega_G = 0.75$. We use the same losses as in StyleGAN-T (Sauer et al., 2023) for discriminator, and a GAN loss as in Esser et al. (2021). We also adopt the adaptive weight $\lambda$ for GAN loss proposed in Esser et al. (2021) to balance the scales of reconstruction and adversarial losses. $\lambda$ is defined as:

$$\lambda = \frac{\|\nabla_{\hat{x}} \mathcal{L}_{rec}\|}{\|\nabla_{\hat{x}} \text{GAN}(\hat{x}, x)\| + \epsilon},$$

**Augmentations.** For data augmentation, we first resize the input image to $384 \times 384$ and then randomly crop to $256 \times 256$. We also apply differentiable augmentations with default hyperparameters in Zhao et al. (2020) before discriminator.

### C.3 VISUALIZATIONS

We present visualizations of reconstructions from different RAEs. As shown in Figure 8, all RAEs achieve satisfactory reconstruction fidelity.

## D DIFFUSION MODEL IMPLEMENTATION

**Datasets.** We primarily use ImageNet-1K (Russakovsky et al., 2015) for diffusion training. Most experiments are conducted at a resolution of $256 \times 256$. For 512-resolution synthesis without decoder upsampling, we train diffusion models directly on $512 \times 512$ images.

**Models.** By default, we use LightningDiT (Yao et al., 2025) as the backbone of our diffusion model. We use a continuous time formulation of flow matching and restrict the timestep input to real values in $[0, 1]$. Following prior work (Song et al., 2021), we replace the timestep embedding with a Gaussian Fourier embedding layer. We also add Absolute Positional Embeddings (APE) to the input tokens in addition to RoPE, though we do not observe significant performance difference with or without APE.

| Model Config | Dim | Num-Heads | Depth |
|---|---|---|---|
| S | 384 | 6 | 12 |
| B | 768 | 12 | 12 |
| L | 1024 | 16 | 24 |
| XL | 1152 | 16 | 28 |
| XXL | 1280 | 16 | 32 |
| H | 1536 | 16 | 32 |
| G | 2048 | 16 | 40 |
| T | 2688 | 21 | 40 |

Table 13: Model configurations for different sizes.

For DiT$^{\text{DH}}$, we generally follow the same architecture as DiT, and does not reapply APE for the DDT head input. We use a linear layer to map the DiT$^{\text{DH}}$ encoder output to the DiT$^{\text{DH}}$ decoder

dimension when the dimension of DiT and DDT head mismatches. We provide a detailed model configuration in Table 13.

**Compute.** For all models based on RAE, we use a patch size of 1. For baseline experiments on VAE and pixel inputs, we use patch sizes of 2 and 16, respectively. Across all $256 \times 256$ experiments, the diffusion model processes a token sequence of length 256. Consequently, the computational cost of the DiT backbone remains identical across RAE, VAE, and pixel settings, with differences arising only from the patchification step. Since patchification accounts for less than 1% of the total GFLOPs, training DiT on different autoencoders introduces only a negligible compute overhead.

**Optimization.** For DiT, we strictly follow the optimization strategy in LightningDiT (Yao et al., 2025), using AdamW with a constant learning rate of $2.0 \times 10^{-4}$, a batch size of 1024 and an EMA weight of 0.9999. We do not observe instability or abnormal training dynamics with this recipe on DiT. For DiT$^{DH}$, we find using the recipe in (Yao et al., 2025) leads to loss spikes at later epochs and slow EMA model convergence at early epochs. We instead use a linear decay from $2.0 \times 10^{-4}$ to $2.0 \times 10^{-5}$ with a constant warmup of 40 epochs. To encourage the convergence of EMA model, we change the EMA weight from 0.9999 to 0.9995. Additionally, we use gradient clipping of 1.0 for DiT$^{DH}$. Other optimization hyperparameters are the same as DiT. All models are trained for 80 epochs unless otherwise specified. We only report EMA model performance.

**Sampling.** We use standard ODE sampling with Euler sampler and 50 steps by default. We find the performance generally converges above 50 steps. We use the same sampling hyperparameters for both DiT and DiT$^{DH}$.

**Computation.** We use PyTorch/XLA on TPU for all training and inference on RAE. For evaluation, we use a single v6e-8 to generate 50K samples. For the 800 epoch DiT$^{DH}$-XL (DINOv2-B) result, we conduct the generation on a machine with 4 A100s and evaluate FID on CPU due to a lack of TensorFlow GPU support. We use an internal JAX codebase for training baseline models on VAEs.

## E    SAMPLING DETAILS FOR FID EVALUATION

For the FID-50K evaluation on RAE, we followed the protocol used in  (Tian et al., 2024; Li et al., 2024b; Ren et al., 2025; Wang et al., 2025c), sampling 50 images from each class for a total of 50,000 images. The reference statistics were taken from the ADM pre-computed statistics (Dhariwal & Nichol, 2021) over the full ImageNet dataset. Another commonly adopted sampling strategy is to uniformly sample class labels 50K times and generate images accordingly (Peebles & Xie, 2023; Ma et al., 2024; Yu et al., 2025). It is worth noting that this strategy is not equivalent to the per-class sampling scheme–although random sampling asymptotically converges to the per-class version, the resulting FID scores differ slightly in practice.

To ensure a fair comparison, we re-evaluate previous state-of-the-art methods that did not use class-balanced sampling with class-balanced sampling. We also evaluate our method with random sampling. As shown in Table 14, all methods' FID improves consistently with class-balanced sampling.

We also note that the original ImageNet training set is inherently unbalanced, with class sizes ranging from approximately 732 to 1,300 samples (Russakovsky et al., 2015). Therefore, the common assumption behind both balanced and random sampling—namely, that the label distribution is uniform—is not entirely accurate. However, since 895 classes contain exactly 1,300 samples, the dataset exhibits a high degree of near-equivalence among most classes. This partial balance may partly explain why balanced sampling consistently yields better results, as it more closely approximates the true label distribution of the training set. As the FID values approach the lower ranges, these subtle details begin to have a greater impact. We therefore want to raise awareness within the community. Finally, while the FID metric remains a useful indicator of generative quality, its absolute value becomes less meaningful as generation fidelity continues to improve.

## F    THEORY EXPERIMENT SETUP

In this section we list the setup of experiments in Section 3 for overfitting images.

| Method | Epochs | Generation@256 w/o guidance | | | | Generation@256 w/ guidance | | | |
|---|---|---|---|---|---|---|---|---|---|
| | | Random | | Balanced | | Random | | Balanced | |
| | | gFID↓ | IS↑ | gFID | IS | gFID↓ | IS↑ | gFID | IS |
| *Autoregressive* | | | | | | | | | |
| VAR (Tian et al., 2024) | 350 | - | - | 1.92 | **323.1** | - | - | 1.73 | **350.2** |
| MAR (Li et al., 2024b) | 800 | - | - | 2.35 | 227.8 | - | - | 1.55 | 303.7 |
| xAR-H (Ren et al., 2025) | 800 | - | - | - | - | - | - | 1.24 | 301.6 |
| *Latent Diffusion with VAE* | | | | | | | | | |
| SiT (Ma et al., 2024) | 1400 | 8.61 | 131.7 | 8.54 | 132.0 | 2.06 | 270.3 | 1.95 | 259.5 |
| REPA (Yu et al., 2025) | 800 | 5.90 | 157.8 | 5.78 | 158.3 | 1.42 | 305.7 | 1.29 | 306.3 |
| DDT (Wang et al., 2025c) | 400 | - | - | 6.27 | 154.7 | 1.40 | 303.6 | 1.26 | 310.6 |
| REPA-E (Leng et al., 2025) | 800 | 1.83 | 217.3 | 1.70 | 217.3 | 1.26 | 314.9 | 1.15 | 304.0 |
| *Latent Diffusion with RAE (Ours)* | | | | | | | | | |
| DiT$^{DH}$-XL (DINOv2-B) | 800 | 1.60 | 242.7 | **1.51** | 242.9 | 1.28 | 262.9 | **1.13** | 262.6 |

Table 14: **Performance of different methods using different sampling strategies.** The officially reported numbers are marked in gray.

**Models.** By default, we use a DiT with depth 12, width 768 and a attention head of 4. The depth varies in $\{384.512, 640, 768, 896\}$ and width varies in $\{4, 12, 16, 24\}$ in Figure 3. Other configurations are the same as Appendix D.

**Targets.** We use three images for overfitting experiments, and all numbers reported are the average on three independent run on each images. We resize all targets to $256 \times 256$ and not use any data augmentation .

**Optimizations & Sampling.** For a single target image, the batch size only influences the timestep. We therefore use a relatively small batch size of 32 and a constant learning rate of $2 \times 10^{-4}$, optimized with AdamW ($\beta = (0.9, 0.95)$). The model is trained for 1200 steps without EMA. For sampling, We use standard ODE sampling with Euler sampler and 25 steps by default.

# G  ADDITIONAL ABLATION STUDIES

## G.1  GENERATION PERFORMANCE ACROSS ENCODERS

As shown in Table 15a, DINOv2-B achieves the best overall performance. MAE performs substantially worse in generation, despite yielding much lower rFID. This shows that a low rFID does not necessarily imply a good image tokenizer. Therefore, we use DINOv2-B as the default encoder for our image generation experiments.

(a) gFID and rFID of different encoders w/ and w/o noisy-robust decoding.

| Model | gFID | rFID |
|---|---|---|
| DINOv2-B | 4.81 / 4.28 | 0.49 / 0.57 |
| SigLIP2-B | 6.69 / 4.93 | 0.53 / 0.82 |
| MAE-B | 16.14 / 8.38 | 0.16 / 0.28 |

(b) gFID and rFID of different DINOv2 sizes w/ and w/o noisy-robust decoding.

| Model | gFID | rFID |
|---|---|---|
| S | 3.83 / 3.50 | 0.52 / 0.64 |
| B | 4.81 / 4.28 | 0.49 / 0.57 |
| L | 6.77 / 6.09 | 0.52 / 0.59 |

(c) Scaling $\tau$ for DINOv2-B.

| $\tau$ | gFID | rFID |
|---|---|---|
| 0.0 | 4.81 | 0.49 |
| 0.5 | 4.39 | 0.54 |
| 0.8 | 4.28 | 0.57 |
| 1.0 | 4.20 | 0.60 |

Table 15: **Ablations on noise-augmented decoder training.** Despite minor drop in rFID, the noise-augmented training strategy can greatly improve the gFID across different encoders and model sizes. Default setups are marked in gray.

| Depth | Width | GFLops | FID ↓ |
|---|---|---|---|
| 6 | 1152 (XL) | 25.65 | 2.36 |
| 4 | 2048 (G) | 53.14 | 2.31 |
| 2 | 2048 (G) | 26.78 | 2.16 |

Table 16: **DDT head should be wide and shallow.** A wide, shallow head yields lower FID than deeper (4-layer G) or narrower (6-layer XL) ones .

| | 2-768 | 2-1536 | 2-2048 | 2-2688 |
|---|---|---|---|---|
| Dino-S | 2.66 | 2.47 | 2.42 | 2.43 |
| Dino-B | 2.49 | 2.24 | 2.16 | 2.22 |
| Dino-L | N/A | 2.95 | 2.73 | 2.64 |

Table 17: **Unguided gFID of different RAE and DDT head.** Larger RAE benefits more from wider DDT head. $d$-$w$: a DDT head with $d$ layers and width $w$. Default setups are marked in gray.

### G.2 DESIGN CHOICES FOR NOISE-AUGMENTED DECODING

We first analyze how noise-robust decoding affects reconstruction and generation. Table 15c shows that larger $\tau$ improves generative FID (gFID) consistently, but slightly worsens reconstruction FID (rFID). This supports our intuition: noise encourages the decoder to learn smoother mappings that generalize better to imperfect latents, improving generation quality, but reducing exact reconstruction accuracy.

To test the robustness of this trade-off, we evaluate different encoders (Table 15a) with $\sigma = 0.8$. Across all encoders, noisy training improves gFID while mildly harming rFID. The effect is strongest for weaker encoders such as MAE-B, where gFID improves from 16.14 to 8.38. Finally, Table 15b shows that the benefit holds across encoder sizes, suggesting that robust decoder training is broadly applicable.

Together, these results highlight a general principle: decoders should not only reconstruct clean latents, but also handle their noisy neighborhoods. This simple change enables RAEs to serve as stronger backbones for diffusion models.

### G.3 DESIGN CHOICES FOR THE DDT HEAD

We now investigate design variants of the DDT head to identify those that serve its role more effectively. Two factors turn out to be crucial: (a) the head needs to be wide and shallow, and (b) its benefit depends on the size of the underlying RAE encoder.

**Width and Depth.** We first vary the architecture of the DDT head, sweeping both width and depth while keeping the total parameter count approximately fixed. As shown in Table 16, a 2-layer, 2048-dim (G) head outperforms a 6-layer, 1152-dim (XL) head by a large margin, despite having similar GFlops. Moreover, a 4-layer, 2048-dim head does not improve over the 2-layer version, even though it has double the GFlops. This suggests that a wide and shallow head is more effective for denoising.

**Dependence on Encoder Size.** Next, we analyze how the effect of the DDT head scales with the size of the RAE encoder. We fix the DiT backbone as DiT-XL and vary the DDT head width from 768 (B) to 1536 (H), 2048 (G), and 2688 (T). We train DiT$^{DH}$ models on top of three RAEs: DINOv2-S, DINOv2-B, and DINOv2-L. As shown in Table 17, the optimal DDT head width increases as the encoder scales. When using DINOv2-S and DINOv2-B, the performance converges at a DDT head width of 2048 (G), while 2688 (T) head still brings performance gains on DINOv2-L. This suggests that the larger RAE encoders benefit more from a wider DDT head.

By default, we use a 2-layer, 2048-dim DDT head for all DiT$^{DH}$ models.

## H ADDITIONAL SCALING RESULTS

We examine how model scale affects training loss in Figure 9. Increasing the model's computational capacity leads DiT$^{DH}$ to converge faster and reach a lower final loss.

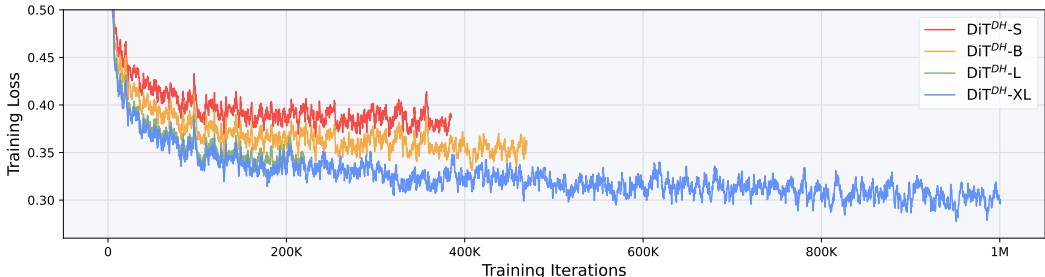

Figure 9: **Training loss of DiT$^{\text{DH}}$ on DINOv2-B.** We use an EMA weight of 0.9 to smooth the loss.

## I   GUIDANCE

We primarily adopt AutoGuidance (Karras et al., 2025) as our guidance method, as it is easier to tune than CFG with interval (Ho & Salimans, 2022; Kynkäänniemi et al., 2024) and consistently delivers better performance. CFG with interval is used only for the DiT-XL + DINOv2-S with guidance result reported in Table 8.

**AutoGuidance.** We adopt AutoGuidance (Karras et al., 2025) as our primary guidance method. The idea is to use a weaker diffusion model to guide a stronger one, analogous to the principle of Classifier-Free Guidance (CFG) (Ho & Salimans, 2022). Empirically, we observe that weaker base models and earlier training checkpoints consistently yield stronger guidance effects. In practice, we use the smallest variant, DiT$^{\text{DH}}$-S, as the guiding model, typically adopting an early checkpoint. Unless otherwise specified, AutoGuidance results are obtained using a guidance scale of 1.5 and a 20-epoch checkpoint of DiT$^{\text{DH}}$-S. For the best result of 1.13 FID (DiT$^{\text{DH}}$-XL on DINOv2-B, $256 \times 256$), we sweep the guidance scale and use a guidance scale of 1.42 with a 14-epoch DiT$^{\text{DH}}$-S checkpoint. Notably, training this base model requires only about 0.05% of the compute used to train the guided model (DiT$^{\text{DH}}$-XL for 800 epochs).

**Classifier-Free Guidance.** We also evaluate CFG (Ho & Salimans, 2022) on RAE. Interestingly, CFG without interval does not improve FID; in fact, applying it from the first diffusion step increases FID. With Guidance Interval (Kynkäänniemi et al., 2024), CFG can achieve competitive FID after careful grid search over scale and interval. However, on our final model (DiT$^{\text{DH}}$-XL with DINOv2-B), the best CFG result remains inferior to AutoGuidance. Considering both performance and tuning overhead, we adopt AutoGuidance as our default guidance method.

## J   DESCRIPTIONS FOR FLOW-BASED MODELS

Diffusion Models (Ho et al., 2020; Dhariwal & Nichol, 2021; Karras et al., 2022) and more generally flow-based models (Albergo et al., 2023; Lipman et al., 2023; Liu et al., 2023) are a family of generative models that learn to reverse a reference "noising" process. One of the most commonly used "noising" process is the linear interpolation between i.i.d Gaussian noise and clean data (Esser et al., 2024; Ma et al., 2024):

$$\mathbf{x}_t = (1 - t)\mathbf{x} + t\boldsymbol{\varepsilon}$$

where $\mathbf{x} \sim p(\mathbf{x}), \boldsymbol{\varepsilon} \sim \mathcal{N}(0, \mathbf{I}_n), t \in [0, 1]$, and we denote $\mathbf{x}_t$'s distribution as $\rho_t(\mathbf{x})$ with $\rho_0 = p(\mathbf{x})$ and $\rho_1 = \mathcal{N}(0, \mathbf{I})$. Generation then starts at $t = 1$ with pure noise, and simulates some differential equation to progressively denoise the sample to a clean one. Specifically for flow-based models, the differential equations (an ordinary differential equation (ODE) or a stochastic differential equation (SDE)) are formulated through an underlying velocity $v(\mathbf{x}_t, t)$ and a score function $s(\mathbf{x}_t, t)$

$$\text{ODE} \qquad \mathrm{d}\mathbf{x}_t = v(\mathbf{x}_t, t)\mathrm{d}t$$

$$\text{SDE} \qquad \mathrm{d}\mathbf{x}_t = v(\mathbf{x}_t, t)\mathrm{d}t - \frac{1}{2}w_t s(\mathbf{x}_t, t)\mathrm{d}t + \sqrt{w_t}\mathrm{d}\bar{\mathbf{w}}_t$$

where $w_t$ is any scalar-valued continuous function (Ma et al., 2024), and $\bar{\mathbf{w}}_t$ is the reverse-time Wiener process. The velocity $v(\mathbf{x}_t, t)$ is represented as a conditional expectation

$$v(\mathbf{x}_t, t) = \mathbb{E}[\dot{\mathbf{x}}_t | \mathbf{x}_t] = \mathbb{E}[\boldsymbol{\varepsilon} - \mathbf{x} | \mathbf{x}_t]$$

and can be approximated with model $v_\theta$ by minimizing the following training objective

$$\mathcal{L}_{\text{velocity}}(\theta) = \int_0^1 \mathbb{E}_{\mathbf{x}, \boldsymbol{\varepsilon}} \left[ \| v_\theta(\mathbf{x}_t, t) - (\boldsymbol{\varepsilon} - \mathbf{x}) \|^2 \right] \mathrm{d}t$$

The score function $s(\mathbf{x}_t, t)$ is also represented as a conditional expectation

$$s(\mathbf{x}_t, t) = -\frac{1}{t} \mathbb{E}[\boldsymbol{\varepsilon} | \mathbf{x}_t]$$

Notably, $s$ is equivalent to $v$ up to constant factor (Albergo et al., 2023), so it's enough to estimate only one of the two vectors.

## K    EVALUATION DETAILS

### K.1    EVALUATION

We strictly follow the setup and use the same reference batches of ADM (Dhariwal & Nichol, 2021) for evaluation, following their official implementation.[3] We use TPUs for generating 50k samples and use one single NVIDIA A100 80GB GPU for evaluation.

In what follows, we explain the main concept of metrics that we used for the evaluation.

- **FID** (Heusel et al., 2017) evaluates the distance between the feature distributions of real and generated images. It relies on the Inception-v3 network (Szegedy et al., 2016) and assumes both distributions follow multivariate Gaussians.
- **IS** (Salimans et al., 2016) also uses Inception-v3, but evaluates logits directly. It measures the KL divergence between the marginal label distribution and the conditional label distribution after softmax normalization.
- **Precision and recall** (Kynkäänniemi et al., 2019) follow their standard definitions: precision reflects the fraction of generated images that appear realistic, while recall reflects the portion of the training data manifold covered by generated samples.

## L    UNCONDITIONAL GENERATION

We are also interested in how RAEs perform in unconditional generation. To evaluate this, we train DiT$^{\text{DH}}$-XL on RAE latents without labels. Following RCG (Li et al., 2024a), we set labels to null during training and use the same null label at generation time. While classifier-free guidance (CFG) does not apply in this setting, AutoGuidance remains applicable. We therefore train DiT$^{\text{DH}}$-XL for 200 epochs with AG (detailed in Appendix I).

As shown in Table 18, our model achieves substantially better performance than DiT-XL trained on VAE latents. Compared to RCG, a method specifically designed for unconditional generation, our approach attains competitive performance while being much simpler and more straightforward, without the need for two-stage generation.

| Method | gFID ↓ | IS ↑ |
|---|---|---|
| DiT-XL + VAE | 30.68 | 32.73 |
| DiT$^{\text{DH}}$-XL + DINOv2-B (w/ AG) | 4.96 | 123.12 |
| RCG + DiT-XL | 4.89 | 143.2 |

Table 18: **Comparison of unconditional generation on ImageNet 256 × 256.**

---

[3] https://github.com/openai/guided-diffusion/tree/main/evaluations

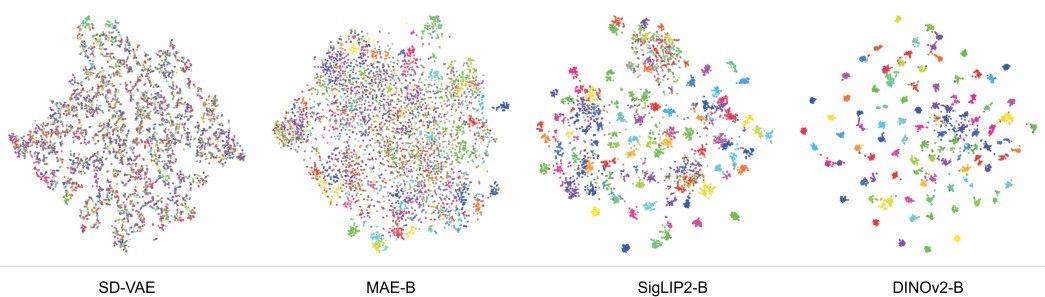

| SD-VAE | MAE-B | SigLIP2-B | DINOv2-B |

Figure 10: **tSNE examples.** We use 50 images per class from ImageNet validation set, and 100 classes for tSNE visualization.

## M    INTERPRETING THE DIFFUSABILITY OF RAE VIA TSNE

A key remaining question is how the choice of RAE encoder influences generation performance. While we demonstrated in Appendix C.1 that the encoder choice dramatically impacts results, the underlying mechanism requires further explanation. We perform tSNE visualizations on the representations from various RAE encoders, using SD-VAE as a baseline. To extract features for visualization, we use the [CLS] token for encoders that possess one (MAE, DINOv2), and apply global pooling over all patch tokens for those that do not (SigLIP2, SD-VAE).

As illustrated in Figure 10, we observe a clear hierarchy in class separation: SD-VAE $<$ MAE $<$ SigLIP2 $<$ DINOv2. DINOv2 exhibits the most distinct clustering among classes, while MAE shows the least separation among the discriminative encoders, though it still surpasses SD-VAE. Intuitively, well-separated class clusters simplify the learning of the generative model; the class-conditioned velocity becomes more coherent within classes and distinct between them, making the diffusion process easier to fit.

## N    VISUAL RESULTS

We show uncurated $512 \times 512$ samples sampled from our most performant model: DiT$^{\text{DH}}$-XL on DINOv2-B with autoguidance scale = 1.5.

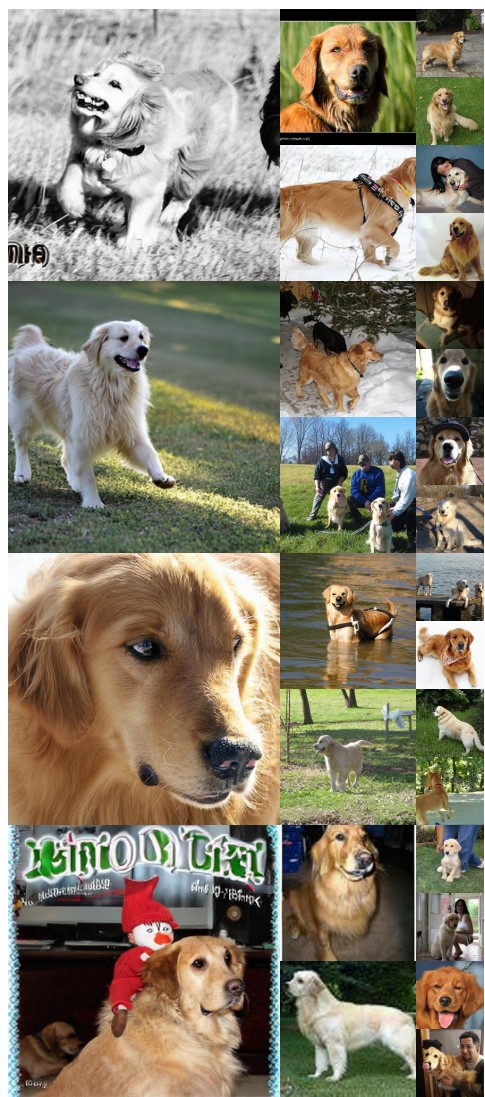

Figure 11: **Uncurated** $512 \times 512$ **DiT$^{\text{DH}}$-XL samples.**
AutoGudance Scale = 1.5
Class label = "golden retriever" (207)

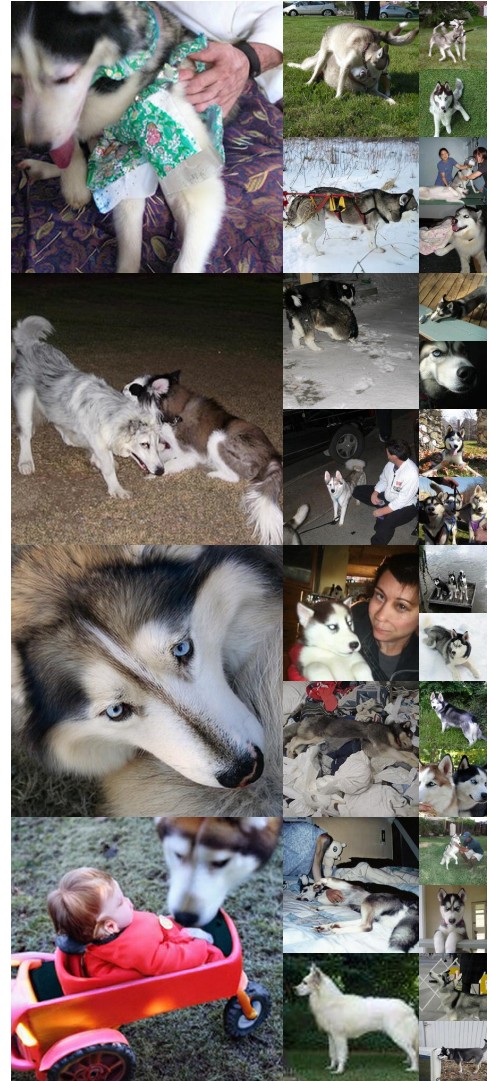

Figure 12: **Uncurated** $512 \times 512$ **DiT$^{\text{DH}}$-XL samples.**
AutoGudance Scale = 1.5
Class label = "husky" (250)

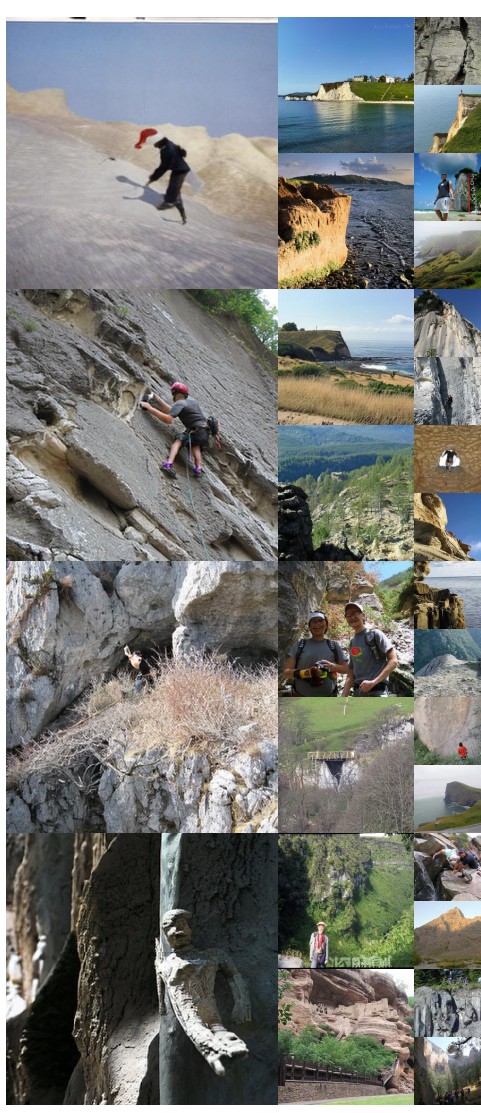

Figure 13: **Uncurated** $512 \times 512$ **DiT$^{\text{DH}}$-XL samples.**
AutoGudance Scale = 1.5
Class label = "cliff" (972)

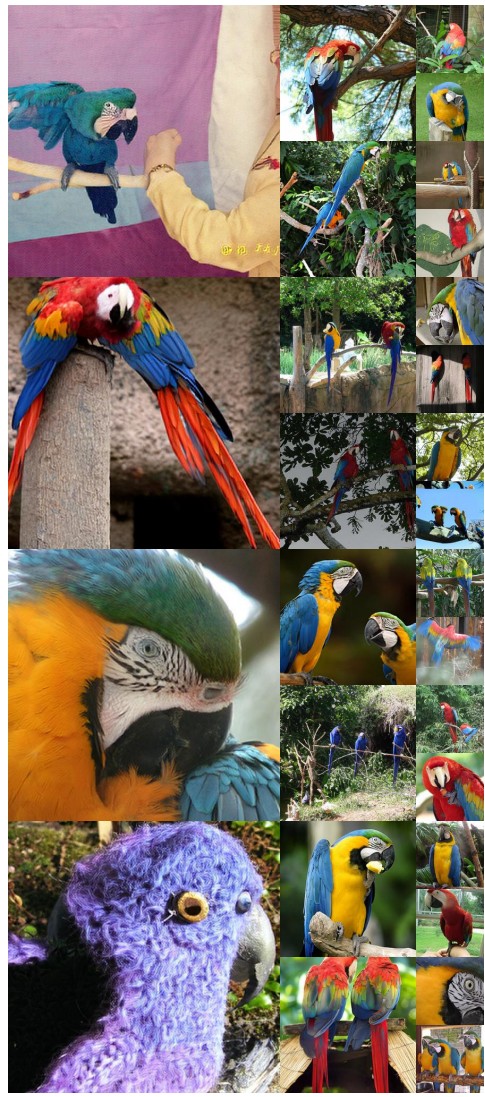

Figure 14: **Uncurated** $512 \times 512$ **DiT$^{\text{DH}}$-XL samples.**
AutoGudance Scale = 1.5
Class label = "macaw" (88)

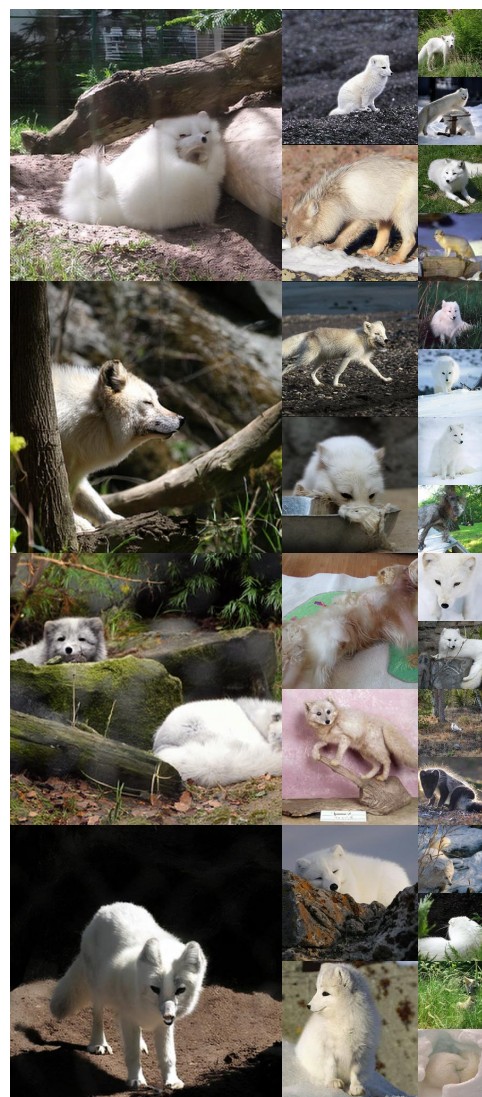 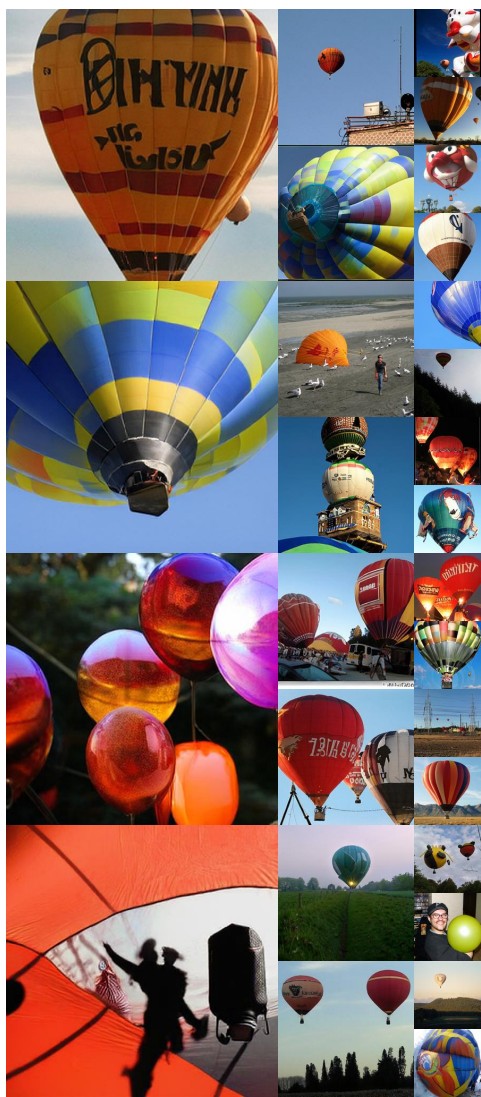

Figure 15: **Uncurated** $512 \times 512$ **DiT$^{\text{DH}}$-XL samples.**
AutoGudance Scale = 1.5
Class label = "arctic fox" (279)

Figure 16: **Uncurated** $512 \times 512$ **DiT$^{\text{DH}}$-XL samples.**
AutoGudance Scale = 1.5
Class label = "balloon" (417)

