# OpenReview forum: "Diffusion Transformers with Representation Autoencoders"
_ICLR.cc/2026/Conference — ICLR 2026 Poster_

### Official Review · Reviewer_UBC1 · 2025-10-27

**Soundness:** 2
**Presentation:** 4
**Contribution:** 3
**Rating:** 4
**Confidence:** 4

**Summary:**

The paper introduced Represenation Autoencoders, a new paradigm for choosing the latent encoder and decoder while training diffusion transformers. Rather than the conventionally used autoencoder SD-VAE, the authors claim that utilizing a preexisting foundation model as the encoder is more suited for training diffusion models since they contain semantic representation as well as portray excellent reconstruction ability. With the newly proposed autoencoder and additional design choices, specifically, (a higher dimensinonal latent space for the transfomer, utilizing a DDT head at the end of the diffusion transformer, a revised noise schedule to suit for Dinov2 latent space) the authors achieve state-of-the-art results in 256x256 and 512x512 generation in ImageNet 1M dataset. With the proposed approach additionally, the authors obtain a representation that seems to be cfg robust.

**Strengths:**

1. The introduced methods, along with the design choices, significantly reduce the number of convergence epochs of training diffusion transformers in the ImageNet 1M dataset.
2. The proposed method seems to be robust with repect to cfg and achieves similar performance with and without cfg. This may enable faster generation with less compute overhead. I would advise the authors to consider this angle while presenting this work as a model that is robust to different cfg values as well as portray good performance without cfg would be useful to the community.
3. The authors have tried extensive experiments to make the DiT model work with Dino-v2  based representation encoder on ImageNet dataset.
4. Finally the paper is well written and each design choice is clearly explained.

**Weaknesses:**

1. The authors have done a good job presenting the rFID numbers and extensive experiments for Dinov2, however, Dinov2 is known to discard information relevant for perfect reconstruction during the encoding process. (Ex:- In Fig 9 , on the bus, all the text-based information is lost, moreover, the headlight bulbs fused into one, and the lines in the bus got joined or bent ) Given that such information cannot be captured by the encoder, how can a generation process be guided by such features? Moreover, looking at Fig 13, 14, 15, there are multiple cases like this where the generated images lose semantic details.
2. How easy is it for RAE to generalize for text-to-image generation? Dinov2 is known to not preserve colours, textual information and straight lines. Would this impact the performance of RAE and its generalizability towards text-to-image generation?
3. Dinov2 features are not discriminative towards instances of the same object class, for example:  if two people are present in the scene, Dinov2 would give similar features for the hands of the two people. Assuming the text prompt is "raise the left hand for the person on the left" would this cause a problem towards the model being able  to identify this with text since all four hands in this case would be given similar features? Some examples can be seen in Figure 11/13 where multiple parrots/ dogs fuse to one.
4. The SDVAE was originally trained for LAION dataset, and the size of the encoder and decoder was made to obtain the best performance in LAION. How does RAE autoencoder perform in LAION? Is a bigger encoder-decoder needed for RAE to work with the LAOIN dataset? As an actionable item, I would like the author's clarification of the performance RAE encoder-decoder on LAION, specifically on the PSNR, SSIM and rFID
5.  There is a performance oscillation for  DiT-DH-XL with different versions of Dinov2 encoders as portrayed in Table 6, and they don't scale linearly with parameter sizes. Why is this? Would this mean that the models may not scale with larger foundation model encoders in the future?
6. In Ln 448-450, the authors claim that "the diffusion models trained for 256x256 could be generalized to 512x512" is this claim valid for other datasets than ImageNet where more number of semantic objects are involved in the same scene?
7. I personally applaud the authors efforts in trying to make the model work with Dino-v2 latent space, however a concern I have is , is such kind of extensive encoder-dependent tuning necessary if we are to swap in a different encoder? Wouldn't this reduce the utility of the approach?

**Questions:**

1. What is the compute in GFLOPS due to the extra 163 M parameters introduced in RAE from Table 6 in RAE? Does this exceed the SDVAE compute cost?
2. Figure 8a and b are a bit confusing; the convergence scores at 10^11 training iterations is different for Dit-DH-XL, Could the authors please clarify this?
3.  Figure 2 portrays that the encoder of RAE has a compute cost of 22GFlops. However, the authors have not clarified which Dinov2 version this refers to. Could the authors please clarify this?
4. Could the authors provide the performance for Table 7 DiT-DH-XL at 20 and 80 epochs in the presence of cfg?
5. Can a smaller encoder-decoder be trained for SDVAE so that it is tuned for ImageNet 1M. Would such an encoder present better reconstruction performance with smaller number of parameters?
6. By the design choices, would RAE need a big diffusion model and a large dino v2 encoder to train diffusion models on smaller datasets like FFHQ?
7. In Table 9, Upsample is spelled wrong.
8. While training  diffusion transformers for smaller datasets like FFHQ wouldn't the limitation of large latent space hold and wouln't the model be severely overparametrized? Additionally, if the method is not generalizable towards text to image generation and infeasible towards smaller datasets, wouldn't this limit the scope of the work?

---

> ### Author Response · Authors · 2025-11-24
> **Response to Reviewer UBC1 (P1)**
>
> We thank the reviewer for the detailed and thoughtful review. We would first like to note that RAE is **not limited to DINOv2**. In fact, it works across all encoders we experimented with, including **DINOv2, SigLIP2, and MAE**. While DINOv2 offers the best generative performance, even the weakest-performing setup—DiT on **MAE**—achieves **FID 8.28 at 80 epochs**, already outperforming baselines such as SiT at 1400 epochs (8.61). Therefore, although DINOv2 may have limitations in its representations (as the reviewer claimed in [W1,W3]), we believe these do **not** affect the general scope or validity of RAE.
>
> We respond to each of the reviewer's comments one-by-one in what follows.
>
> [W1] : **Imperfect DINOv2 reconstruction**
>
> We understand the reviewer’s concern about imperfect reconstruction quality in DINOv2. However, we would like to clarify that **imperfect reconstruction does not necessarily mean the encoder has lost the underlying information**. The reconstruction quality depends jointly on the encoder and the **decoder**, and a decoder that is undertrained or exposed to insufficient data may fail to recover details that the encoder still preserves.
>
> To validate this, we conduct additional experiments where the **RAE decoder is trained on more data** (details in Appendix. N.1). As shown in below, scaling the decoder training leads to consistent improvements in **rFID**, **PSNR**, and **SSIM** on both ImageNet and a subset of YFCC web images. We further provide qualitative comparisons in the [image link](https://freeimage.host/i/fKfcuOF) (also Fig. 14), which show that training on more diverse data significantly improves reconstruction quality—particularly for text regions and small objects. This demonstrates that many of the details initially thought to be “lost” can in fact be recovered with a sufficiently trained decoder.
> | Family | Model        | rFID (IN) ↓ | PSNR (IN) ↑ | SSIM (IN) ↑ | rFID (YFCC) ↓ | PSNR (YFCC) ↑ | SSIM (YFCC) ↑ |
> |--------|-----------------------|-------|--------|---------|-------|--------|---------|
> | RAE    | SigLIP2-So (IN)       | 0.462 | 20.82 | 0.563 | 0.970 | 20.95 | 0.591 |
> | RAE    | SigLIP2-So (Web)      | **0.435** | **21.34** | **0.593** | **0.702** | **21.75** | **0.628** |
>
> (IN) indicates that the decoder is trained on ImageNet, while (Web) indicates that the decoder is trained on web images.
>
> Additionally, **worse reconstruction does not necessarily imply worse generation**. In fact, the opposite is often true:
> - **VAE:** Increasing the VAE bottleneck improves reconstruction but worsens generation [A].
> - **RAE:** MAE achieves the strongest reconstruction (rFID 0.28) yet yields the weakest generation among our encoders.
> - **Pixel:** Pixel representations have perfect reconstruction by definition, yet pixel-space diffusion is known to be more difficult than latent diffusion.
>
> Furthermore, we show that **adding mild noise to the decoder** during training slightly harms reconstruction but *improves* generation quality (Sec. 4.3), further illustrating that reconstruction and generation capture different properties of the representation.
>
> We hope this clarification addresses the reviewer’s concern about the relationship between reconstruction fidelity and generative performance.
>
>
> [W2]: **Generalizability to T2I**
>
> We appreciate the reviewer’s interest in the generalizability of RAE to text-to-image generation. We have worked hard to demonstrate that RAE extends beyond class-conditional generation, and while the work is still ongoing, our preliminary results already provide strong evidence that the advantages of RAE over VAE carry over to the T2I setting. Under the same experimental setup (described in Appendix N), RAE outperforms FLUX-VAE on standard T2I benchmarks such as **GenEval** and **DPG-Bench**, and achieves a **3.75× faster** convergence rate ([image link]( https://freeimage.host/i/fKfcIHP), Fig. 15). These results suggest that RAE remains a promising and competitive alternative in the T2I regime.
> | Gen.Tok.   | Steps | GenEval | DPG  |
> |-----------|--------|---------|------|
> | FLUX-VAE  | 30k    | 39.6    | 70.5 |
> | RAE       | 8k     | 40.0    | 72.1 |
> | RAE       | 30k    | 49.5    | 74.2 |

---

> ### Author Response · Authors · 2025-11-24
> **Response to Reviewer UBC1 (P2)**
>
> [W3] : **Possible prompt misalignment due to similar features**
>
> We understand the reviewer’s concern regarding potential prompt–image alignment issues. Training a model to perform **image editing** — as the reviewer requested,  `raise the left hand for the person on the left`—typically requires a dedicated post-training pipeline and specialized image-editing datasets, which are beyond the scope of our current work.
>
>
> As an alternative, we provide multiple **text-to-image** examples using similar prompts of the form: *“Generate an image of an `X` on the left of another. The `X` on the right/left [does something].”* This setup explicitly tests whether the model can distinguish **similar features** between two nearly identical objects.
>
> As shown in the [image link](https://freeimage.host/i/fKfcTR1) (Fig. 16), the model produces correct images across all cases, demonstrating that it can reliably identify the two entities and follow the prompt accordingly. We hope this additional experiment resolves the reviewer’s concern.
>
> [W4]  : **SD-VAE is optimized for LAION**
>
> We understand the reviewer’s concern that SD-VAE may perform better if more carefully tuned on ImageNet. However, we respectfully disagree. The commonly used SD-VAE model (`sd-vae-ft-ema`) is finetuned from `sd-vae-kl-f8` (as clarified in the official [model card](https://huggingface.co/stabilityai/sd-vae-ft-ema)), which itself was optimized specifically for ImageNet rFID (“no further substantial improvement in rFID”, per the [official documentation](https://github.com/CompVis/latent-diffusion#pretrained-autoencoding-models)). After finetuning on LAION, the ImageNet rFID further improves from **0.90 → 0.62**, indicating that LAION-based fine-tuning actually benefits ImageNet reconstruction.
>
> Additionally, SD-VAE is trained on an **unreleased, curated subset** of LAION, making a strictly fair comparison impossible. To make relatively fair comparisons, we extended RAE decoder training to **more diverse web images** and evaluated both RAE and SD-VAE on a **50k disjoint set** of YFCC web images. As shown below, although RAE achieves lower PSNR/SSIM, it shows a **substantial advantage in rFID** compared to SD-VAE. Qualitative examples in the [image link](https://freeimage.host/i/fKfcuOF) (Fig. 14) further highlight RAE’s stronger reconstruction of fine details—particularly **text regions**, where SD-VAE often fails.
> | Family | Model        | rFID (IN) ↓ | PSNR (IN) ↑ | SSIM (IN) ↑ | rFID (YFCC) ↓ | PSNR (YFCC) ↑ | SSIM (YFCC) ↑ |
> |--------|--------------|-------------|-------------|-------------|----------------|----------------|----------------|
> | VAE    | SD-VAE       | 0.978       | 24.78       | 0.705       | 0.987          | 25.25          | 0.738          |
> | RAE    | DINOv2-L*    | 0.388       | 22.18       | 0.637       | 0.556          | 22.52          | 0.669          |
> | RAE    | SigLIP2-So     | 0.435 | 21.34 | 0.593 | 0.702 | 21.75 | 0.628 |
>
> (* indicate we use WebSSL[B], a variant of DINOv2.)

---

> > ### Author Response · Authors · 2025-11-24
> > **Response to Reviewer UBC1 (P3)**
> >
> > [W5]  : **Encoder scaling**
> >
> > We thank the reviewer for raising the topic of encoder scaling. It is true that RAE with **DINOv2-L** performs worse than with **DINOv2-B**, but this phenomenon is not unique to our method. Prior work that leverages DINO representations for generative modeling—such as **REPA** and **REPA-E**—also observes that DINOv2-B outperforms DINOv2-L. A recent [ICLR submission](https://openreview.net/forum?id=y0UxFtXqXf) further suggests that the underlying cause may be related to **spatial correlation properties** of the encoder rather than model size.
> >
> > These consistent observations indicate that a larger encoder is not necessarily a *better* encoder for generation, and this behavior should not be interpreted as a limitation of RAE’s scalability. Instead, it highlights an important gap in current representation learning practice. We encourage the community to consider **generation performance** as an additional evaluation protocol when developing and assessing visual representations.
> >
> >
> > [W6] : **Generalizability of decoder upsampling**
> >
> > We understand the concern regarding the generalizability of decoder upsampling. To address this, we trained an upsampling decoder (224 → 378) within our text-to-image model, whose training data contain **multiple semantic objects**. As shown in [image link](https://freeimage.host/i/fKfcTR1) (Fig. 16), the decoder continues to perform well in these multi-object scenarios, demonstrating that decoder upsampling remains effective beyond simple object-centric cases.
> >
> > [W7] **Encoder-dependent tuning**
> >
> > We would like to clarify that **we do not tune the diffusion model or the decoder training specifically for DINOv2**. Our diffusion recipe and decoder recipe are applied uniformly across **all** encoders we evaluate, **without tuning any hyperparameters**. As shown in Tab. 14(a), even the weakest-performing encoder—**MAE**—achieves **gFID 8.23 at 80 epochs**, outperforming SiT at 1400 epochs (8.61), representing more than a **10× speedup**. Our effort focuses on adapting the VAE-oriented diffusion recipe to RAE’s high-dimensional semantic latent space, rather than only adapting to DINOv2. Using the *same* recipe, we successfully train decoders and DiTs on SigLIP2, MAE, and DINOv2, all of which yield strong reconstruction and generation performance.

---

> ### Author Response · Authors · 2025-11-24
> **Response to Reviewer UBC1 (P4)**
>
> [Q1]: **Additional GFlops**
>
> To quantify the impact of the additional 163M parameters in RAE, we report a detailed GFLOPs breakdown (also in Tab. 18):
>
> | Method                | AE GFLOPs (256) | Diff GFLOPs (256) | gFID (256) | AE GFLOPs (512) | Diff GFLOPs (512) | gFID (512) |
> |-----------------------|------------------|---------------------|------------|------------------|--------------------|------------|
> | **SiT**               | 445.87           | 118.64              | 2.06       | 1783.49          | 524.60             | 2.62       |
> | **REPA**              | 445.87           | 118.64              | 1.42       | 1783.49          | 524.60             | 2.08       |
> | **DDT**               | 445.87           | 118.64              | 1.26       | 1783.49          | 508.46             | 1.25       |
> | **DiTDH-XL (DINOv2-B)** | 129.02           | 145.03              | 1.18       | 513.57           | 638.83             | 1.13       |
>
> The extra 163M parameters in RAE correspond to the **DiTDH-XL (DINOv2-B)** configuration. The extra parameters brings 26.38 additional GFlops to the diffusion model at 256 resolution. However, RAE has hundreds GFlops less than SD-VAE. Therefore, the total (AE + diffusion) compute cost does not exceed that of SD-VAE–based methods. A recent follow-up, MF-RAE[B], exploits this advantage to build fast and performant consistency models on RAE.
>
>
> [Q2]: **Inconsistency in Fig. 7**
>
> We thank the reviewer for pointing this out. The confusion stems from a mislabeled caption. The correct caption for Fig. 7(a) should be **“Model FLOPs”** rather than “Training GFLOPs.”
>
> - **Fig. 7(a)** compares models with different **per-forward FLOPs** at the **same number of training epochs**, so the x-axis reflects *model FLOPs* rather than total training GFLOPs.
> - **Fig. 7(b)**, in contrast, plots performance as a function of **training GFLOPs**.
>
> We've corrected the caption in the revision. Thanks again for pointing out!
>
>
> [Q3]: **Version of DINO**
>
> We apologize for the lack of clarity. The **22 GFLOPs** shown for the RAE encoder in Fig. 2 correspond to **DINOv2-B**, which is also our main and most performant configuration throughout the paper.

---

> ### Author Response · Authors · 2025-11-24
> **Response to Reviewer UBC1 (P5)**
>
> [Q4]: **Performance with guidance**
>
> We agree that reporting results with guidance provides a more complete comparison. We would like to clarify that **our main results use AutoGuidance (AG)** rather than classifier-free guidance (CFG). AG is designed to be applied **at later training stages**, as it uses an *under-trained model* as the base guidance model. This is why guidance results of early epochs are not shown in the main tables.
>
> As requested by the reviewer, we still evaluate DiT-DH-XL under our default AG setup and report:
>
> | Epoch | w/o Guidance | w/ AG |
> |-------|--------------|-------|
> | 20    | 3.71         | 3.31  |
> | 80    | 2.16         | 1.59  |
>
> As shown, AG provides more substantial gains at later epochs, consistent with its intended usage. We hope this additional results addresses the reviewer’s request.
>
>
> [Q5]: **Smaller SD-VAE tuned for ImageNet-1k**
>
> Due to time and computational constraints, we were not able to re-design and retrain a smaller SD-VAE specifically tailored for ImageNet-1k, as there is no mature prior practice to follow. However, as discussed in [W4], **training on more diverse images** (e.g., LAION-style data) tends to **improve** SD-VAE’s reconstruction performance on ImageNet. For example, the rFID improves from **0.90 → 0.62** after LAION fine-tuning. This suggests that restricting training solely to ImageNet-1k is unlikely to yield a *better* ImageNet-optimized SD-VAE compared to one trained on a larger and more diverse corpus. We hope this addresses the reviewer’s concern.
>
>
>
> [Q6]: **Over-parameterization on small datasets and need for large models**
>
> We appreciate the reviewer’s concern about potential over-parameterization on smaller datasets such as FFHQ. We clarify the following:
>
> 1. **Encoder size.** Our default encoder, **DINOv2-B**, is already **much lighter than SD-VAE encoder**, with roughly **3× fewer GFLOPs**. Additionally, smaller variants such as **DINOv2-S** also work well within our framework (Tab. 7), indicating that RAE does **not** inherently require a large DINOv2 model.
>
> 2. **Diffusion model size.** RAE is fully compatible with **small diffusion models**. Our smallest variant, **DiTDH-S**, requires only ~**33 GFLOPs** (similar to DiT-B), yet it **outperforms several VAE baselines that rely on DiT-XL** (120 GFlops), as shown in Fig. 6.\(c\). This demonstrates that RAE improves efficiency rather than forcing the use of larger models.
>
> Regarding FFHQ: we agree that evaluating on smaller datasets would be valuable. Unfortunately, most DiT-based baselines we compare against do not report FFHQ, making direct comparison difficult. For this reason, we focus on ImageNet and text-to-image settings where strong, well-established baselines exist.
>
>
>
> [Q7]: **Typo in Table 9**
>
> We thank the reviewer for catching this. We have corrected the typo “Upsample” in Table 9 in the revised manuscript.
>
> [Q8]: **Scope, generalizability, and small datasets**
>
> We thank the reviewer for raising this point. As discussed in [Q6], training diffusion models with RAE does **not** require either a large encoder or a very large diffusion model —smaller DINO variants and DiTDH-S work well and remain compute-efficient.
>
> Regarding text-to-image generation, as shown in [W2] and [W3], we extend RAE to **T2I settings** and demonstrate that:
>
> - RAE-based T2I models **outperform FLUX-VAE** under the same setup on GenEval and DPG-Bench,
> - RAE achieves a **3.75× speedup in convergence**,
>
> Taken together, these results indicate that RAE is **not restricted** to ImageNet-level generation. It scales to T2I generation, and it can be instantiated with smaller encoders and diffusion models, which we hope alleviates the reviewer’s concerns about the scope and generalizability of the approach.
>
> [A]: Yao et al., Reconstruction vs. Generation: Taming Optimization Dilemma in Latent Diffusion Models (2025)
>
> [B]: Fan et al., Scaling Language-Free Visual Representation Learning (2025)
>
> [C]: Hu et al., MeanFlow Transformers with Representation Autoencoders (2025)

---

### Official Review · Reviewer_qR4u · 2025-10-29

**Soundness:** 3
**Presentation:** 4
**Contribution:** 3
**Rating:** 6
**Confidence:** 4

**Summary:**

This paper discusses how to replace the VAE encoder in LDMs with pretrained representation encoders (e.g., DINOv2). The authors conduct a thorough analysis around this topic, and the paper is well-written, clear, and provides meaningful insights.

**Strengths:**

1. Introducing semantic tokenizers into generative diffusion models is a valuable and timely exploration. This idea may also be extended to unified MLLMs to alleviate the representational conflict between generation and understanding tasks caused by different semantic and pixel-level granularity requirements.

2. The paper is highly readable and the proposed method is elegant. The analyses on model width, noise schedule, and the proposed DDT head provide clear insights and further improve the effectiveness of semantic tokenizers for generation tasks.

3. Extensive experiments and analyses demonstrate that the proposed method achieves SOTA performance.

**Weaknesses:**

1. The results in Table 1 require further clarification.  RAEs with frozen encoders achieve consistently better reconstruction quality (rFID) than SD-VAE (lines 156–159). However, concurrent works [1, 2] reported that semantic tokenizers tend to decrease reconstruction fidelity.  Additional explanations for why rFID improves, along with more experiments, could help further clarify this point.

2. DINOv2 was trained on hundreds of millions of images [3], which may include ImageNet-like, object-centric samples. This raises potential concerns about evaluation fairness. It would be helpful if the authors could clarify how to avoid the data leakage risk.

3. The notably low gFID on ImageNet may raise concerns about potential overfitting. Since incorporating semantic latents could simplify the latent structure [4], a (large) pretrained tokenizer might more easily memorize simplified semantic distributions. Further OOD evaluations would help confirm the generality of the improvement.

4. It would be interesting to explore whether a smaller-width DiT, combined with a simple learnable linear module, could achieve comparable performance of the d>n setting. For example, consider the model $s_{\theta}(x_t) = Wx_t + Ug(V^Tx_t)$ where $V \in \mathbb{R}^{n \times r}$ and $r \ll n$. According to Theorem 4.2 in [5], the oracle score can be expressed as a linear term with a low-rank correction. This could potentially help alleviate the need for a large model width.

5. It would be interesting to discuss whether such semantic tokenizers could be incorporated into unified MLLMs (e.g., for joint generation, understanding, and editing) to mitigate representational conflicts between different tasks [6].

[1] Vision Foundation Models as Effective Visual Tokenizers for Autoregressive Generation.

[2] Aligning Visual Foundation Encoders to Tokenizers for Diffusion Models.

[3] DINOv2: Learning Robust Visual Features without Supervision.

[4] Masked Autoencoders Are Effective Tokenizers for Diffusion Models.

[5] Can Diffusion Models Learn Hidden Inter-Feature Rules Behind Images?

[6] Janus-Pro: Unified Multimodal Understanding and Generation with Data and Model Scaling.

**Questions:**

See Weaknesses.

---

> ### Author Response · Authors · 2025-11-24
> **Response to Reviewer qR4u (P1)**
>
> We deeply appreciate the reviewer's insightful comments and efforts in reviewing our manuscript. We respond to each of the reviewer's comments one-by-one in what follows.
>
> [W1] **Why rFID improves?**
>
> We thank the reviewer for raising this important point. The key difference between our approach and concurrent works [A, B] lies in how the semantic representations are treated before reconstruction.
> Although [A, B] also rely on pretrained visual encoders, they apply **substantial compression or quantization** to the high-dimensional continuous features, which inevitably harms reconstruction fidelity. Specifically:
> - [A] **quantizes** continuous embeddings into *12-dimensional discrete codes*, leading to substantial information loss.
> - [B] introduces an **adapter** that compresses high-dimensional encoder features into a *32-dimensional latent* to make generation easier, again sacrificing spatial and frequency detail.
>
> In contrast, RAE trains a decoder **directly** on the *full-dimension, continuous* latent space of the encoder, without any quantization or dimensionality reduction. This preserves substantially more visual information, allowing the RAE decoder to achieve *higher* reconstruction fidelity (lower rFID). We believe this distinction is the core reason why RAE outperforms these semantic tokenizers in reconstruction, and it highlights a fundamental difference between RAE and [A, B].
>
>
> [W2] **Data leakage concern**
>
> We appreciate the reviewer’s concern regarding potential data leakage in DINOv2 pretraining. We address this from three angles:
>
> **(1) Strong performance without DINOv2.**
>
> RAE does not rely on DINOv2 to achieve strong results. For instance, **RAE with SigLIP2** achieves **FID 4.93** at 80 epochs—close to DINOv2 (4.28) and already outperforming the majority of VAE baselines (for example, REPA: 7.90). SigLIP2 is trained on large-scale, diverse web images and is not explicitly optimized for ImageNet-style distributions, mitigating concerns about dataset overlap. We also note that the SD-VAE baseline is likewise trained on web-scale data.
>
>
> **(2) Fair comparison with VAE baselines.**
>
> The three strongest VAE baselines we compare against—**DDT**, **REPA**, and **REPA-E**—also incorporate DINOv2 features during training. Since these baselines already leverage DINOv2, we believe that evaluating RAE with DINOv2 is a **fair and consistent** comparison.
>
> **(3) Additional T2I experiments using SigLIP2.**
>
> To further rule out leakage concerns, we conducted text-to-image experiments using **SigLIP2** instead of DINOv2. Under the same setup (detailed in Appendix N.2), RAE **outperforms FLUX-VAE** on GenEval and DPG-Bench and achieves a **3.75× faster** convergence rate ([image link]( https://freeimage.host/i/fKfcIHP), Fig. 15). These results reinforce that the advantages of RAE over VAE persist on large scale data.
> | Gen.Tok.   | Steps | GenEval | DPG  |
> |-----------|--------|---------|------|
> | FLUX-VAE  | 30k    | 39.6    | 70.5 |
> | RAE       | 8k     | 40.0    | 72.1 |
> | RAE       | 30k    | 49.5    | 74.2 |
>
> We hope these additional experiments and clarifications help address the reviewer’s concerns regarding data leakage.
>
>
>
> [W3] **Overfitting concern and OOD evaluation**
>
> To address the reviewer’s concern about potential overfitting or reliance on simplified semantic structures, we evaluate **out-of-training-distribution performance** using the ImageNet **validation** split, which is disjoint from the training set. We take the generated images with lowest reported train FID for evaluation. As shown below, RAE **outperforms** the previous strongest VAE-based baselines (REPA-E) indicating better coverage of the real data distribution:
>
> | Method  | val FID  | Prec. | Rec. |
> |---------|------|-------|--------|
> | REPA-E  | 2.48 | **0.70**  | 0.66   |
> | RAE     | **2.14** | 0.69  | **0.67**   |
>
> These results provide additional evidence that the gains from RAE generalize beyond the training distribution rather than arising from overfitting.

---

> ### Author Response · Authors · 2025-11-24
> **Response to Reviewer qR4u (P2)**
>
> [W4] **Additional linear skip connection $W$**
>
> We thank the reviewer for the insightful suggestion. In theory, adding a simple full-rank learnable linear skip-connection does satisfy the width requirement in a parameter-efficient manner. This aligns well with the formulation in Theorem 4.2 of [C], where the oracle score admits a decomposition into a linear term plus a low-rank correction.
>
> In practice, we observe that inserting such a linear module indeed improves performance over the vanilla DiT-S baseline (results shown below). However, a **single linear layer** has limited expressive power for capturing the score function of realistic, high-dimensional data distributions. This motivates our design of the **DDT head**, which can be viewed as an enhanced version of the proposed skip-connection: instead of a single linear transformation, we employ **two lightweight transformer blocks**, providing richer representation capacity, more robust score estimation, and consistently stronger empirical performance.
>
> We would also like to highlight that this design is closely related to **JiT** [D], a recent method that explores pixel-space diffusion with DiTs with x-prediction. JiT can be interpreted as applying a skip-linear correction (−\(x_t\) + model velocity prediction), followed by a fixed \(1/(1-t)\) scaling—conceptually similar to the linear pathway suggested by the reviewer. We additionally conduct experiments with JiT and find that it significantly boosts performance, improving the FID of DiT-S from 80 → 28.
>
> We really appreciate the reviewer’s suggestion and will make this connection explicit in the revision.
>
> | Model            | Plain DiT-S | DiT-S w/ $W$ | DiT-S w/ JiT | DiT-S w/ DDT Head|
> |------------------|-------------|--------------|--------------|-------------- |
> | FID              | 215.76      | 80.50        | 28.28        | 6.07          |
>
>
>
>
> [W5] **Unified Models**
>
> We agree that evaluating RAE in the context of unified multimodal models (UMMs) is an exciting direction. In our text-to-image experiments (detailed in Appendix N), we already adopt a **single shared vision encoder**—the RAE encoder—for both understanding and generation, thereby removing the need for dual encoders commonly used in existing UMM architectures. This demonstrates that RAE naturally supports the “unified latent space” design philosophy highlighted by the reviewer.
>
> Following the reviewer’s suggestion, we additionally evaluate the **image understanding** capability of the model in Appendix N.3 (Tab. 20). Compared with an understanding-only baseline, incorporating generative training data **does not degrade** the model’s visual understanding performance. This indicates that RAE is compatible with unified modeling: the same latent space can support both **high-quality generation** and **strong understanding**.
>
> | Model       | MMEₚ   | TVQA | AI2D | Seed | MMMU | MMMUₚ |
> |-------------|--------|------|------|------|------|--------|
> | Und.-only   | 1374.8 | 44.7 | 63.9 | 67.1 | 40.2 | 20.5   |
> | RAE-based   | 1468.7 | 39.6 | 66.7 | 69.8 | 41.1 | 19.8   |
>
>
>
> Reference:
> [A]: Zheng et al., Vision Foundation Models as Effective Visual Tokenizers for Autoregressive Image Generation (2025)
>
> [B]: Chen et al., Aligning Visual Foundation Encoders to Tokenizers for Diffusion Models (2025)
>
> [C]: Han et al., Can Diffusion Models Learn Hidden Inter-Feature Rules Behind Images? (2025)
>
> [D]: Li T., He K., Back to Basics: Let Denoising Generative Models Denoise (2025)

---

### Official Review · Reviewer_diB6 · 2025-11-01

**Soundness:** 3
**Presentation:** 3
**Contribution:** 3
**Rating:** 8
**Confidence:** 3

**Summary:**

This paper proposes a new replacement to VAE autoencoder (used for downstream Diffusion Transformer models) with Representation Autoencoder (RAE). The authors argue that the traditional UNET based VAEs are computationally heavy and provide a weak latent space that leads to lower performance. Their RAE takes a pretrained representation encoder (like DINOv2) and lightweight ViT-based decoder to solve this issue. This new Autoencoder is not directly applicable to the DiT architectures 1) The image reconstruction is limited, hence a wide DiT is used 2) noise schedule shift is employed to account for increased dimensions 3) The latents are augmented with noise to inject stochasticity. To scale however, a shallow DDT head is attached to the DiT backbone which achieves FID scores of 1.18 on ImageNet-256

**Strengths:**

1)  The proposed shallow DiT + RAE method achieves an FID of 1.18 on ImageNet 256x256 and 1.13 on 512x512, outperforming all prior diffusion models.

2) The method converges faster than VAE-based baselines  for example 10x speedup over VA-VAE while maintaining the FID scores.

3) The RAE also employs an efficient decoder. This part also contributes to make the model faster.

4) The paper provides a theoretical proof and empirical evidence, that the DiT width should be at least equal to the dimensionality of the tokens.

**Weaknesses:**

1) The paper does not necessarily prove that the better reconstruction quality equals to better generation quality. This still raises questions on the diffusibility of the the latent space proposed in the paper.

2) The model is suitable for the high computation models, for the highly compressed models in VAE which can have very high compression ratios, this model might not be comparable in the computation cost do to the requirement of dimensionality and DiT width constraints.

3) Again Shallow DiT only benefits high dimension latents , for example it can perform worse that DiT-XL trained on low dim SD-VAE.

4) The new training scheme is unstable as compared to  the standard DiT which is stable with a constant learning rate.

**Questions:**

1) The fundamental question is what property beyond reconstruction makes a representation diffusable?  and why does DINOv2 succeed where MAE fails?

2) What is the qualitative trade-off of noise-augmented decoding, if the FID improves but is the reconstruction compromised?

---

> ### Author Response · Authors · 2025-11-24
> **Response to Reviewer diB6 (P1)**
>
> We deeply appreciate the reviewer's insightful comments and efforts in reviewing our manuscript. We respond to each of the reviewer's comments one-by-one in what follows.
>
> [W1, Q1]: **Diffusability of representation**
>
> We appreciate the reviewer’s insightful question on what makes a representation *diffusible*. While this remains an open research problem, we provide an analysis in the Appendix (t-SNE visualization) via tSNE.
>
> As shown in the [visualization](https://freeimage.host/i/fKfcYiv) (Fig. 17), A clear hierarchy emerges:
> **SD-VAE < MAE < SigLIP2 < DINOv2** in terms of class separability.  DINOv2 exhibits the most distinct class clusters, whereas MAE shows the weakest separation among discriminative encoders (though still stronger than SD-VAE). Intuitively, well-separated class manifolds simplify the diffusion learning problem: the class-conditioned velocity field becomes more coherent within each class and more distinguishable across classes. This correlates with the observed generation ranking (**MAE < SigLIP2 < DINOv2**). We present this as an initial characterization rather than a definitive explanation. A full understanding of how encoder representation influences diffusion training is an open research question deserving further investigation.
>
> We also note that **better reconstruction does not imply better generation**. In fact, the opposite is often true:
> - VAE: Increasing the VAE bottleneck improves reconstruction but worsens generation [A].
> - RAE: MAE achieves the strongest reconstruction (rFID 0.28) yet yields the weakest generation among our encoders.
> - Pixel has perfect reconstruction but are known to be harder for diffusion models than it's latent counterpart.
>
> Additionally, we show that **adding mild noise to the decoder** during training slightly hurts reconstruction but *improves* generation quality (Sec. 4.3), illustrating again that these metrics target different aspects of the representation.
>
> Reconstruction should therefore be viewed as a *proxy* objective — one that implicitly assumes the diffusion model can perfectly “memorize” the training samples. In practice, diffusion models are known to **underfit** [B] and generate novel samples, breaking this assumption. As a result, better reconstruction does not necessarily imply better generation performance. We hope this clarification, together with the t-SNE analysis, addresses the reviewer’s concern.
>
>
> [Q2]: **Qualitative tradeoff of noise-augmented decoding**
>
> We agree with the reviewer that qualitative examples of noise-augmented decoding are helpful. We have added visualizations at [image link](https://freeimage.host/i/fKfcADg) (Fig. 18). As shown in these examples, noise-augmented decoding produces images with stronger high-frequency details and improved IS/FID, despite having worse reconstruction fidelity. This offers additional evidence that reconstruction quality and generation quality measure different properties of the latent representation, and that better reconstruction does not imply better generation.
>
>
> [W2] **Compression ratio & dimension constraints**
>
> (a) **Compression Ratio.**
> We agree with the reviewer that VAEs can indeed achieve higher compression ratios—for instance, DC-AE [C] compresses 512-resolution images into 256 tokens, whereas our naïve 512-resolution RAE setting uses 1024 tokens (matching DiT, SiT, and REPA). However, RAE can be made equally or more efficient. As shown in Sec. 6.1, the diffusion model can operate on **256-resolution latents**, and the decoder upsamples to 512. This reduces the token count to **256 tokens**, identical to DC-AE, while achieving **better generative performance**. Specifically, DC-AE reports FID 1.72 with a 2B-parameter UViT, whereas our RAE achieves **FID 1.62** with a significantly smaller 893M-parameter DiT-DH-XL.
>
> (b) **Dimension Constraints.**
> The purpose of introducing DiT-DH is precisely to address the dimensionality constraint and bring parameter efficiency to diffusion on RAE latents. As illustrated in Fig. 8.\(c\), our smallest model, **DiT-DH-S**, uses only **33 GFLOPs** (similar to DiT-B) while outperforming several VAE-based baselines that depend on **DiT-XL (120 GFLOPs)**. Thus, we respectfully disagree that RAE is only suitable for high-compute models—RAE is also highly effective in low-compute regimes.

---

> > ### Author Response · Authors · 2025-11-24
> > **Response to Reviewer diB6 (P2)**
> >
> > [W3] **Shallow DiT doesn’t benefit low-dimensional VAE**
> >
> > We agree with the reviewer that DiT-DH primarily benefits **high-dimensional** latent spaces. The goal of DiTDH is not to universally improve DiT on all tokenizers; rather, it is designed as an adaptation of DiT that addresses the computational and optimization challenges unique to **high-dimensional** inputs such as RAE latents. As shown in Fig. 8.\(b\), this adaptation yields substantially faster convergence and stronger final performance on RAE, underscoring its importance for RAE-based diffusion models. Additionally, we show in Sec. 6.3 that DiTDH also beneifits high-dimensional pixel inputs, indicating DiTDH as a general improvement on general high-dimension inputs.
> >
> > [W4] **Unstable training**
> >
> > We respectfully disagree that DiTDH exhibits unstable training. DiTDH adds only two standard components relative to vanilla DiT—**learning-rate decay** and **gradient clipping**—both widely used in LLM training [D,E] and diffusion models [F,G]. We additionally provide the training loss curve in Fig. 19 (Appendix P), which shows smooth behavior with no spikes or divergence. This confirms that DiT-DH training is stable in practice.
> >
> > [A]: Yao et al., Reconstruction vs. Generation: Taming Optimization Dilemma in Latent Diffusion Models (2025)
> >
> > [B]: Song et al., Selective Underfitting in Diffusion Models (2025)
> >
> > [C]: Chen et al., Deep Compression Autoencoder for Efficient High-Resolution Diffusion Models (2024)
> >
> > [D]: Yang et al., Qwen2.5 Technical Report (2024)
> >
> > [E]: Touvron et al., Llama 2: Open Foundation and Fine-Tuned Chat Models (2023)
> >
> > [F]: Liang et al., Scaling Laws for Diffusion Transformers (2024)
> >
> > [G]: Tang et al., Exploring the Deep Fusion of Large Language Models and Diffusion Transformers for Text-to-Image Synthesis (2025)

---

### Official Review · Reviewer_tJTk · 2025-11-05

**Soundness:** 3
**Presentation:** 2
**Contribution:** 3
**Rating:** 6
**Confidence:** 4

**Summary:**

This paper revisits the training paradigm of image tokenizers for image generation models. It addresses the limitations of the standard SD-VAE–style tokenizer, whose training primarily depends on reconstruction losses and therefore often yields latent spaces with limited semantic structure. The authors propose an unconventional approach: replacing the learnable encoder with a frozen, pretrained vision encoder and training only the decoder for image reconstruction. To mitigate the resulting challenges—such as a substantially higher latent dimensionality and a more fragile decoder—the paper introduces several architectural and training refinements to the DiT models. The resulting tokenizer demonstrates improved image generation quality and faster convergence.

**Strengths:**

- This paper targets a very important problem of rich semantics in image tokenization. The proposed tokenizer design is simple yet very effective.

- The practical and theoretical justifications for the architectural design are very convincing.

- The empirical results are strong on ImageNet-level evaluations.

**Weaknesses:**

**Presentation**

My major concern with this paper is that it seems to have been submitted in a huge rush. A throughout proofreading and reformatting is very necessary. Examples:

- I failed to find Figure 3 and Figure 4 in the main manuscript.

- Line 175: the two sentences beginning with “Because RAE…” are repetitive.

- Line 345: missing space before 'To'.

- Figure 8 appears earlier than Figure 7 in the manuscript.


**Technical**

- In Line 013 of the abstract. I personally believe it is not appropriate to classify traditional VAEs as U-Nets. U-Nets typically include skip connections between the encoder and decoder, which are not applicable in image tokenization models.

- In Line 086 of Introduction. Calling the latent space of RAE 'discrete' can be very misleading. Readers might confuse it with typical latent space such as VQ. If the authors wish to retain this term, a clear explanation is required early in the paper; otherwise, readers will remain unclear about its meaning until Section 4.3.

- In line 306, the statement “noise smooths the latent distribution and mitigates OOD issues for the decoder, but also removes fine details” is unconvincing. Since the additional noise is introduced only during training, and the decoder receives clean latents during inference, the argument that noise removes fine details does not hold.

- In Table 9, compared to direct scaling tokens, the upsample method actually increases the gFID by 42%, which can hardly be considered as 'comparable' as stated by the authors. This raises concerns about the scalability of the proposed method.

**Questions:**

- I understand that the authors aim to ensure fair comparisons by controlling the model size in terms of parameter count. However, having the same number of parameters does not necessarily imply comparable efficiency across all aspects. Could the authors also report the wall-clock training and inference speeds, as well as the FLOPs, of the proposed model relative to the baselines?

- I understand that the rebuttal period is limited, and I'm not requesting significant experiments here. However, for future submissions, including moderate-scale text-to-image experiments would be valuable to more convincingly demonstrate the practical applicability of the proposed method.

---

> ### Author Response · Authors · 2025-11-24
> **Response to Reviewer tJTk (P1)**
>
> We deeply appreciate the reviewer's insightful comments and efforts in reviewing our manuscript. We respond to each of the reviewer's comments one-by-one in what follows.
>
> [W1] : **Presentation**
>
> We sincerely thank the reviewer for highlighting these presentation issues. We have carefully revised the manuscript to address all identified problems, including missing figures, formatting inconsistencies, and repetitive sentences. Modifications are highlighted in blue in the updated version.
>
> [W1-1] : **It is not appropriate to classify traditional VAEs as U-Nets**
>
> We appreciate the reviewer pointing out the term “UNet.” Although SD-VAE internally uses an [implementation](https://github.com/CompVis/taming-transformers/blob/3ba01b241669f5ade541ce990f7650a3b8f65318/taming/models/vqgan.py#L28) identical to UNet *without* skip connections, we agree that this can be misleading. We have replaced the terminology with **“convolutional”** to avoid confusion.
>
> [W1-2] : **Calling the latent space of RAE 'discrete' can be very misleading**
>
> Thank you for noting the ambiguity. We have clarified the phrase to **“latent space with discrete support.”**
>
> [W1-3] : **The argument that noise removes fine details does not hold**
>
> Thank you for pointing out the confusion. Our claim primarily concerns the effect of **adding noise during decoder training**:
>
> 1. The injected noise removes certain high-frequency details from the latents, making the reconstruction task slightly harder—which explains the small degradation in rFID.
> 2. This also introduces a mild train–test mismatch during reconstruction evaluation, further contributing to the observed reconstruction drop.
>
> We emphasize that the purpose of this additive noise is to **improve generation quality**, not reconstruction. During generation, latents produced by the diffusion model inevitably deviate from the clean DINOv2 latent distribution due to discretization and approximation errors [A]. Adding noise during decoder training makes the decoder more robust to this distribution shift. We will revise the paper to make this motivation more explicit.
>
> [W1-4] : **Scalability of the decoder upsampling method**
>
> By “competitive” (L454), we mean competitive with VAE-based baselines such as REPA (FID 2.08). Additionally, since decoder-based upsampling reduces both training and inference compute by **4×**, it is expected that gFID at 512 resolution will increase. To further isolate the compute factor, we train the 256-resolution model for **4× more epochs**, matching the effective compute. This achieves **FID 1.42** (25% higher than the 1.13 baseline) while still using **4× less inference compute**. We hope it better addresses the reviewer’s concern.

---

> ### Author Response · Authors · 2025-11-24
> **Response to Reviewer tJTk (P2)**
>
> [Q1] **GFlops and wall-clock time**
>
> We agree that reporting both GFLOPs and wall-clock time provides a more complete picture of computational efficiency. We have already included a training GFLOPs comparison in Fig. 6(b) in the main paper, where RAE consistently outperforms all baselines by a substantial margin. In addition, we have expanded Appendix M to include detailed breakdowns of **autoencoder GFLOPs**, **diffusion model GFLOPs**, and **overall wall-clock training time**, offering a clearer and more comprehensive comparison. We hope these additions provide a more transparent and thorough understanding of the compute trade-offs.
>
> | Method                      | AE Param. (256) | AE GFLOPs (256) | Diff Param. (256) | Diff GFLOPs (256) | Wall-clock (256) | gFID (256) | AE Param. (512) | AE GFLOPs (512) | Diff Param. (512) | Diff GFLOPs (512) | Wall-clock (512) | gFID (512) |
> |-----------------------------|------------------|-------------------|---------------------|----------------------|--------------------|------------|------------------|-------------------|---------------------|----------------------|--------------------|------------|
> | **SiT**                     | 84M              | 445.87            | 675M               | 118.64              | 8.88 steps/sec     | 2.06       | 84M              | 1783.49           | 675M                | 524.60               | 2.49 steps/sec     | 2.62       |
> | **REPA**                    | 84M              | 445.87            | 675M               | 118.64              | 8.88 steps/sec     | 1.42       | 84M              | 1783.49           | 675M                | 524.60               | 2.49 steps/sec     | 2.08       |
> | **DDT**                     | 84M              | 445.87            | 675M               | 118.64              | 8.12 steps/sec     | 1.26       | 84M              | 1783.49           | 675M                | 508.46               | 2.31 steps/sec     | 1.25       |
> | **DiTDH-XL (DINOv2-B)**       | 501M             | 129.02            | 839M               | 145.03              | 7.20 steps/sec     | 1.18       | 501M             | 513.57            | 839M                | 638.83               | 2.07 steps/sec     | 1.13       |
> | **DiTDH-XL (DINOv2-B, Upsampling)** | –        | –                 | –                   | –                   | –                  | ––         | 503M             | 129.70            | 839M                | 129.02               | 7.20 steps/sec     | 1.61       |
>
>
> [Q2] **T2I experiments**
>
>
> We appreciate the reviewer’s interest in the generalizability of RAE to text-to-image generation. We have worked hard to demonstrate that RAE extends beyond class-conditional generation, and while the work is still ongoing, our preliminary results already provide strong evidence that the advantages of RAE over VAE carry over to the T2I setting. Under the same experimental setup (details described in Appendix. N.2), RAE outperforms FLUX-VAE on standard T2I benchmarks such as **GenEval** and **DPG-Bench**, and achieves a **3.75× faster** convergence rate ([image link](https://freeimage.host/i/fKfcIHP), Fig. 15). These results suggest that RAE remains a promising and competitive alternative in the T2I regime.
>
> | Gen.Tok.   | Steps | GenEval | DPG  |
> |-----------|--------|---------|------|
> | FLUX-VAE  | 30k    | 39.6    | 70.5 |
> | RAE       | 8k     | 40.0    | 72.1 |
> | RAE       | 30k    | 49.5    | 74.2 |
>
> [A]: Xu et al., Restart Sampling for Improving Generative Processes (2023)

---

### Public Comment · ~Ping_He6 · 2025-11-14
**More results about image reconstruction quality of RAE will make this paper sounder.**

As my observation on the usage of RAE autoencoder, the reconstructed images significantly lose fine details of images fed into RAE and produce obvious artifact.

I would like to see more metrics about image reconstruction quality, such as PSNR and SSIM.

---

> ### Author Response · Authors · 2025-11-24
> **Response**
>
> Thank you for your interest in our work. We have extended RAE training to both web-scale images and text-to-image generation. Below we provide reconstruction and generation comparisons (with full details in Appendix N). Although RAE achieves lower PSNR/SSIM, it shows a **substantial advantage in rFID** compared to SD-VAE. Qualitative examples in the [image link](https://freeimage.host/i/fKfcuOF) (Fig. 14) further highlight RAE’s stronger reconstruction of fine details—particularly **text regions**, where SD-VAE often fails.
>
> | Family | Model        | rFID (IN) ↓ | PSNR (IN) ↑ | SSIM (IN) ↑ | rFID (YFCC) ↓ | PSNR (YFCC) ↑ | SSIM (YFCC) ↑ |
> |--------|--------------|-------------|-------------|-------------|----------------|----------------|----------------|
> | VAE    | SD-VAE       | 0.978       | 24.78       | 0.705       | 0.987          | 25.25          | 0.738          |
> | RAE    | DINOv2-L*    | 0.388       | 22.18       | 0.637       | 0.556          | 22.52          | 0.669          |
> | RAE    | SigLIP2-So   | 0.462       | 20.82       | 0.563       | 0.970          | 20.95          | 0.591          |
>
> We also note that **better reconstruction does not imply better generation**. In fact, the opposite is often true:
> - VAE: Increasing the VAE bottleneck improves reconstruction but worsens generation [A].
> - RAE: MAE achieves the strongest reconstruction (rFID 0.28) yet yields the weakest generation among our encoders.
> - Pixel has perfect reconstruction but are known to be harder for diffusion models than it's latent counterpart.
>
> The same conclusion holds for our text-to-image experiments. Under the same experimental setup (details described in Appendix. N.2), RAE outperforms state-of-the-art FLUX-VAE on standard T2I benchmarks such as **GenEval** and **DPG-Bench**, and achieves a **3.75× faster** convergence rate ([image link](https://freeimage.host/i/fKfcIHP), Fig. 15). These results further reinforce that stronger reconstruction does not translate to better generation—even at web-scale beyond ImageNet.
>
> | Gen.Tok.   | Steps | GenEval | DPG  |
> |-----------|--------|---------|------|
> | FLUX-VAE  | 30k    | 39.6    | 70.5 |
> | RAE       | 8k     | 40.0    | 72.1 |
> | RAE       | 30k    | 49.5    | 74.2 |
>
> [A]: Yao et al., Reconstruction vs. Generation: Taming Optimization Dilemma in Latent Diffusion Models (2025)

---

> > ### Public Comment · ~Ping_He6 · 2025-11-25
> >
> > Besides artifacts during image reconstruction, there are also many artifacts produced by DiT$^{DH}$-XL model without the AutoGuidance of additional DiT$^{DH}$-S model in the generation stage.

---

> ### Author Response · Authors · 2025-11-27
> **Response**
>
> Thank you for the comment. We are unsure which specific artifacts the commenter is referring to. In our experiments, we did not observe notable artifacts from DiTDH-XL when trained with RAE latents, and our method consistently outperforms VAE alternatives across **quantitative metrics** as shown in Tab. 7
>
> To better understand and address the concern, could the commenter provide:
> 1) **Quantitative evidence** indicating degradation (e.g., which metric and under which setting),
> 2) **Qualitative examples** of the artifacts observed, and
> 3) **A baseline or competing method** that performs better under the same configuration?
>
> Such details would allow us to investigate the issue more precisely and clarify any misunderstanding.

---

### Author Response · Authors · 2025-11-24
**Common Response**

Dear Reviewers and AC,

We sincerely appreciate the time and effort you have dedicated to reviewing our manuscript.

We propose the Representation Autoencoder (RAE), a new class of autoencoders that directly uses frozen, pretrained representation encoders (e.g., DINO, SigLIP, MAE) paired with lightweight, trainable decoders. As a replacement for VAEs, training Diffusion Transformers (DiTs) on RAEs achieves faster convergence and stronger performance with only simple adaptations. As the reviewers highlighted, the paper is well-written (diB6, qR4u, UBC1), targets a valuable and timely problem (qR4u, tJTk), introduces simple, elegant, and convincing design choices (qR4u, tJTk), and is supported by strong empirical results together with comprehensive experiments and analyses (all).

We greatly appreciate your constructive feedback. In response, we have carefully revised and enhanced the manuscript as follows:

* RAE training on web images (Appendix N.1)
* Text-to-image experiments (Appendix N.2, N.3)
* Detailed compute comparison (Appendix M)
* Decoder upsampling for text-to-image generation (Figure 16)
* Image understanding with RAE (Table 20)
* t-SNE analysis (Appendix O)
* Qualitative samples of noise-augmented decoding and training loss curves (Appendix P)
* Fixed typos

In the revised manuscript, these updates are temporarily highlighted in **blue** for your convenience.

We sincerely hope these updates help better convey the benefits of the proposed RAE to the ICLR community. We look forward to further discussion.

Thank you very much,
Authors.

---

### Meta-Review · Area_Chair_Cd2W · 2026-01-03

**Summary:**

The reviewers generally acknowledge the merits of the work, and the authors' rebuttal has successfully addressed most concerns.
While minor clarifications (e.g., W1–4 of Reviewer tJTk) are still required, the concern raised by Reviewer UBC1 and a public commenter about the observation that generated images lose semantic details in multiple cases remains. This concern should be discussed in the next version. Additionally, the authors are encouraged to incorporate discussions including linear skip connections, and the model's broader generalizability to text-to-image tasks in the next version. Based on the current scores and the authors' rebuttal, I recommend acceptance.

**Reviewer Concerns:**

Reviewer tJTk noted several presentation flaws, and suggested to add wall-clock training and inference speeds and include some text-to-image experiments. Most of these concerns are properly addressed, while W1-4 still requires final clarification in the revised manuscript. (which is a minor point).

Reviewer diB6 requested further clarification about diffusability of representatio, qualitative trade-off of noise-augmented decoding, unstability of new training scheme, and dimension constraints. Most of the concerns have been addressed, and the original score is already high.

Reviewer qR4u requested further clarification on the and the potential for incorporating semantic tokenizers into unified MLLMs, raised concerns regarding data leakage risk and overfitting. The reviewer also talked about additional linear skip connection (which I think is interesting and worth further discussion). Most of the concerns have been addressed.

Reviewer UBC1 raised some concerns related to the use of Dinov2, as well as the generalizability to text-to-image tasks, and requested further clarification of the performance RAE encoder-decoder on LAION, and more detailed reports on computational costs. While many concerns have been addressed, the observation that generated images lose semantic details in multiple cases remains a valid issue—one that I also share and note has been highlighted by public comments. I suggest the authors address this specific concern in the next version of the manuscript.

**Reviewer Scores:**

I think they may keep the score.

---

### Decision · Program_Chairs · 2026-01-26

Accept (Poster)